



# ISBA-MEB (SURFEX v8.1): model snow evaluation for local-scale forest sites

Adrien Napoly[1], Aaron Boone[1], and Theo Welfringer[2]

[1]CNRM-Université de Toulouse, Météo-France/CNRS, Toulouse, France
[2]National School for Water and Environmental Engineering (ENGEES), Strasbourg, France

**Correspondence:** Adrien Napoly (adr.napoly@gmail.com)

**Abstract.** An accurate modeling of the effect of snow cover on the surface energy and mass fluxes is required from land surface models. The Interactions between Soil–Biosphere–Atmosphere (ISBA) model adopts a default configuration using a composite soil-vegetation energy budget approach which is shown to have certain limitations for representing snow and soil phase change processes in areas of high vegetation cover since it does not explicitly represent the snow pack lying on the ground below the canopy. In particular, previous studies using ISBA have pointed out that the snowpack ablation tends to occur to early in the season in forest regions in the northern hemisphere. The multi-energy balance (MEB) version of ISBA has been developed recently, to a large degree, to address this issue. A vegetation layer, which is distinct from the soil, has been added to ISBA and new processes are now explicitly represented such as snow interception and an under-story litter layer. To evaluate the behavior of this new scheme in a cold forested region, long-term offline simulations have been performed for the three Berms forest sites located in Saskatchewan, Canada. It is shown that the new scheme leads to an improved energy budget representation, especially in terms of the ground and sensible heat fluxes, with decreases in RMSE of 77 and 18 %, respectively. A consistent positive impact for soil temperatures is obtained, particularly in terms of bias which is reduced from -6.2 to -0.1 K at a 10 cm soil depth on average for the three sites and 12 studied years. The impact of using MEB on the snowpack simulation is in a better agreement with observations during the snow season, especially in terms of the time of ablation: errors are on the order of 1 day averaged over the 3 sites and all of the years using MEB, which represents a reduction in error of 20 days compared to the composite scheme. The analysis shows that this improvement is mostly cause by the ability of MEB to represent a snowpack that nearly completely covers the soil below the canopy decouples the soil from the atmosphere while keeping a close coupling between the vegetation and the atmosphere.

## 1 Introduction

Forests cover approximately one third of world's land surface area, and one third of which consists in boreal forest which corresponds to subarctic and cold continental climates. In these regions, snowpack can last more than half of the year and can modify the surface roughness, thermal and radiative properties, thereby having a significant impact on the fluxes of momentum, heat and water mass between the surface and the atmosphere or the soil. Vegetation canopy processes in forests modulate the behaviour (accumulation and melting) of the snowpack on the ground. Notably, snowfall can be intercepted by the canopy





leaves and branches where it can be sublimated or melted before unloading to the forest floor (Pomeroy et al., 1998, Storck et al., 2002 Bartlett et al., 2006). In addition, downwelling shortwave radiative fluxes are attenuated by the sheltering effect of the canopy (Harding and Pomeroy, 1996) while the longwave radiation reaching the below-canopy snow surface is generally enhanced compared to its atmospheric component due to longwave radiation emission by the canopy and trunks (Gouttevin et al., 2015; Todt et al., 2018). The snowpack constitutes a very efficient thermal insulating material that decreases the cooling

of the soil compared to a snow-free surface (Zhang, 2005; Grundstein et al., 2005) which in turn can have a significant impact on soil freezing and thawing and thus on the permafrost depth (Stieglitz et al., 2003; Paquin and Sushama, 2015)

Land surface models (LSMs) seek to provide realistic simulations of snow evolution, which implies that they have the ability to represent the previously mentioned first-order processes. An accurate representation of the impact of snow cover on the energy and water balances is required for land surface reanalysis products, particularly in cold regions (Carrera et al., 2015)

and for operational regional scale hydrological modeling for which snow-melt is a key driver of discharge (e.g.s Habets et al. 2008; Snow et al., 2016). In addition, more physically-based multi-layer snow schemes have been developed for operational numerical weather prediction (Dutra et al., 2010) including explicit forest canopy formulations (Yang et al., 2011), and such schemes have also been developed for climate modeling (e.g.s Oleson et al., 2010; Decharme et al., 2016).

The consensus of GCM predictions for the current century is that high latitude regions will continue to warm at an accelerated

rate compared to other regions of the globe in large part owing to the positive snow albedo feedback (Flanner et al., 2011; Qu and Hall, 2014). This mechanism is considered to be a driver of the observed Arctic amplification of the current global warming (e.g.s Bony et al., 2006; Chapin et al., 2005; Serreze and Barry, 2011). But it is known that the spread in surface albedo feedback among different CMIP5 GCMs is particularly large in the boreal forest zone (Qu and Hall, 2014), and this is in large part owing to the representation of snow masking by vegetation (Thackeray et al., 2015). Thackeray et al. (2018) state that the main reason

for the spread in surface albedo is owing to structural aspects of the LSMs (the representation of snowpack, the vegetation canopy and their interactions) rather than the parameter values used by these schemes.

In the 1990s, a series of Model Inter-comparison Projects (MIPs) were initiated in order to inter-compare and evaluate the LSM state of the art representation of cold season processes with the goal of determining which aspects of the schemes were affecting performance and causing model spread, and also to provide guidance for future model developments. Multiple MIPs

at the local scale, for which detailed measurements of snow processes exist, have been done over the past 20 years, such as the Programme for Intercomparison of Land-Surface Parameterization Schemes (PILPS) Phase 2d (Slater et al., 2001), SnowMIP Phase 1 Etchevers et al. (2004) and Phase 2 (Essery et al., 2009; 2009, Rutter et al.). The PILPS-Phase 2e experiment (Bowling et al., 2003) looked at the combined effect of multiple cold season processes at the regional scale over a Scandinavian catchment. The Rhone-AGGregation MIP (Boone et al., 2004) evaluated the snow depth simulations of an emsemble of LSMs at

numerous observation sites in the French Alps: they determined that liquid water retention was a key processes required for simulating the accurate timing and amount of snowmelt and thus discharge in a high alpine catchment. All of the aforementioned MIPs used observation-based forcing as boundary conditions to the LSMs in offline (decoupled from the atmosphere) mode. Most recently, the Earth System Model Snow Model Intercomparison Project (ESM-SnowMIP: Krinner et al., 2018) extended the inter-comparison to the global scale and also in fully coupled GCM-LSM models. Note that in particular, SnowMIP2





and ESM-snowMIP evaluations highlighted the difficulties LSMs have to model snow in forested sites compared to open sites for a large number of LSMs.

In order to represent certain snow processes in forested regions, many LSMs have adopted an explicit representation of the vegetation canopy. In the late 1980s and early 1990s, the so-called two-source energy budget method began to be implemented into GCMs. In this approach, the surface (which can consist in soil or snow, or a blend thereof) is distinct from the overlying

bulk vegetation canopy, each computing their own fluxes and having explicit parameters. The first and most simplified version was proposed by Deardorff (1978). Based on that approach, Sellers et al. (1986) proposed one of the first comprehensive schemes for use in a GCM which inspired and still resembles many of the two-source LSMs in use today. Over time, more variations of this type of approach have emerged, such as LSMs using simplified treatments for certain processes such as radiative transfer and a significantly reduced number of input parameters (more easily adapted for use in GGMs) while still

retaining the overall explicit canopy (Xue et al., 1991). Some LSMs have further split the canopy into two layers (Saux-Picart et al., 2009) representing the over-story and under-story vegetation layers, or by separating the trunk from the vegetation canopy with a focus on including important long-wave radiative impacts which can be critical for below canopy snowpack evolution (Gouttevin et al., 2012). More recently, models have been developed using a multi-layer vegetation canopy for GCM applications (Ryder et al., 2016) with improvements to the explicit treatment of turbulence (Bonan et al., 2018).

The Interaction Soil-Biosphere-Atmosphère (ISBA) LSM is part of the platform SURFEX (SURFace EXternalisée: Externalized Surface) software platform being developed at Météo-France in collaboration with multiple international partners (Masson et al., 2013). SURFEX is used in operational systems such as numerical weather prediction within the global atmospheric model ARPEGE (Action de Recherche Petite Echelle Grande Echelle) operational at Météo France or the limited area model AROME (Seity et al., 2011) or hydrological and land surface analysis systems (Habets et al., 2008). It is also used

within the CNRM-CM6 climate model (Decharme et al., 2019) which is particpating in the Coupled Model Intercomparison Project phase 6 (CMIP6) project (Eyring et al., 2015). Several key updates and improvements have recently been made to ISBA, notably for cold season processes, such as the representation of soil ice, liquid water and temperature over multiple soil layers, an improved explicit snowpack including more layers and improved thermal conductiivty and albedo computaions and the inclusion of soil organics (Decharme et al., 2016). Despite these improvements, ISBA still has difficulties simulating the

snowpack evolution and soil temperatures for forested sites as evidenced by SnowMIP2. In addition, Decharme et al. (2019) identified a precocious snowmelt over boreal forest regions by a global simulation which lead to a springtime peak of river discharge over all Arctic basins which was too early.

For these reasons, the Multi Energy Balance (MEB, Boone et al., 2017) option was recently implemented within ISBA to allow an explicit and distinct representation of the upper ground and vegetation layers. Radiative transfer models for short

and long wave radiation are improved, interception of snow by the canopy layer is added and turbulent fluxes formulation are adapted to the new design. A parameterization has also been added to model the litter on the ground (Napoly et al., 2017) which reduces soil evaporation and heat exchanges with the soil. The MEB option has been evaluated on a large number of local forest sites (Napoly et al., 2017), but little attention has been paid specifically to its impact on the simulation of snow until now.





In the current study, the ability of ISBA-MEB to model the snowpack is evaluated using data from the Boreal Ecosystem Research Study (BERMS) which covers a twelve year (01/01/1999 - 31/12/2010) observational period for three distinct (aspen, jack pine and black spruce dominated) Canadian forest sites (Bartlett et al., 2006). The current operational ISBA (single composite soil-vegetation scheme) is used as a reference model, so that the aim of this study is to highlight the performance of the new MEB option for the modeling of the snowpack and the related variables. After a presentation of the main contrasting characteristics of each model and a description of the forest sites, the two options are evaluated and compared using the data from the Berms sites with a focus on the modeling of certain key features of the snowpack. Sensitivity tests are also summarized, focusing on the most uncertain parameters of the new MEB option.

## 2 Model

The ISBA model is developed within the SURFEX platform (SURace EXternalisée, Masson et al. 2013) and version 8.1 is used in this study. There are multiple parameterization options available, notably those which govern soil thermal and hydrological fluxes, snowpack physics and the new explicit vegetation canopy and forest litter options. Note that ISBA within SURFEX includes the notion of explicit sub-grid patches which represent different types of plant functional types or land classes explicitly, but in the current study we use a single-patch representation (thus throughout the text, we refer to patch or grid cell interchangeably). The options most pertinent to the current study are described below.

### 2.1 ISBA : Default configuration

ISBA uses a so-called composite approach which is defined herein as using a single energy balance for the combined soil-vegetation surface Fig. 1a. The properties of soil and vegetation are aggregated depending on the fraction that vegetation occupies ($veg$) in the considered grid-cell (Noilhan and Planton, 1989 ; Noilhan and Mahfouf, 1996). Since the inception of ISBA, many developments have been made to improve the representation of physical processes as the knowledge of key processes, the quality and spatial and temporal coverage and resolution of input datasets and the computing speed have improved. In this paper, we use the default ISBA configuration for research studies which consists in:

- the soil water and energy transfers are simulated using the diffusive approach option (DIF) (Boone et al., 2000; Decharme et al., 2011) that uses multiple (here 12) layers to solve the Fourier and Darcy laws throughout the soil. The soil parameters are derived from soil texture using pedotransfer functions based on Clapp and Hornberger (1978) classification. The impact of soil organic carbon (SOC) on thermal and hydraulic properties is also used (Decharme et al., 2016)

- the parameterization of the stomatal resistance used to calculate the forest transpiration models the functional coupling between the stomatal resistance and the net assimilation of $CO2$ (Ag-s, Calvet et al., 1998). An option to simulate the evolution of the leaf area index (LAI) prognostically is not activated in the current study since estimated values are available and thus imposed.





– the snowpack is modeled using a multi-layer physically-based explicit snow option (ES) which was first developed by Boone and Etchevers (2001). Since that time, multiple improvements over the ensuing 15 years have been implemented and are described by Decharme et al. (2016). The key physical processes are briefly summarized in Section 2.1.3.

In the following section, we describe the aspects related to snow representation in the model that differ between the default version of ISBA and the new MEB option.

## 2.1.1   Energy budget

The energy budget equations for the composite surface soil-vegetation (hereafter simply referred to as the composite layer) and upper snow layer, are are expressed as follows:

$$\mathcal{C}_s \frac{dT_s}{dt} = (R_{nets} - LE_s - H_s)(1 - p_{sn}) + G_{ns}\,p_{sn} - G_{s,1} + L_f \Phi_{s,1} \tag{1}$$

$$\mathcal{C}_{n,1} \frac{\partial T_{n,1}}{\partial t} = R_{netn} - H_n - LE_n - \tau_{n,1}SW_{netn} + \xi_{n,1} - G_{n,1} + L_f \Phi_{n,1} \tag{2}$$

where $T_s$ (K) is the temperature of the composite surface and $T_{n,1}$ (K) represents the temperature of the uppermost layer of snow. $\mathcal{C}_s$ and $\mathcal{C}_{n,1}$ (J K$^{-1}$ m$^{-2}$) are the effective heat capacities of the composite and upper snow layers, respectively. Both budgets use a relatively thin layer (for soil and snow: on the order of several cm maximum) in order to be able to properly model the surface temperature diurnal cycle. $R_{nets}$ and $R_{netn}$ (W m$^{-2}$) correspond to the net radiative fluxes for the soil and the snowpack, respectively. In the same way, $LE_s$, $LE_n$, $H_s$ and $H_n$ (W m$^{-2}$) are the latent and sensible heat fluxes, respectively.

$G_{s,1}$ and $G_{n,1}$ (W m$^{-2}$) are the conductive fluxes from the composite and snow surface layers to the corresponding sub-surface layers, respectively. The conductive heat flux between the base of the snowpack and the composite layer is represented by $G_{ns}$. The effective heating (or cooling) rate of a snowpack layer caused by exchanges in enthalpy between the surface and sub-surface model layers when the vertical grid is reset (the snow model grid-layer thicknesses vary in time) is represented by $\xi_{n,1}$. The phase change terms (freezing less melting, expressed in kg m$^{-2}$ s$^{-1}$) are represented by $\Phi_s$ and $\Phi_n$ respectively, and $L_f$

represents the latent heat of fusion (J kg$^{-1}$).

    The fraction of the soil-vegetation surface covered by snow is $p_{sn}$, thus the surface soil layer is in contact simultaneously with both the base of the snowpack and the atmosphere when $p_{sn} < 1$ (which represents a critical difference with MEB which will be discussed further in a subsequent section). Also note that the budget in Eq. 2 is snow-relative: in order to obtain the total energy budget and the net fluxes for a patch containing snow, all of the terms in the snow energy budget are multiplied by

$p_{sn}$ and then added to Eq. 1. Several of the terms most critical to cold season processes in Eq.s 1-2 are described in more detail in the following sections.

## 2.1.2   Radiative Transfer

The surface net radiation of the soil-vegetation and snowpack are given by

$$R_{net} = p_{sn}(SW_{netn} + LW_{netn}) + (1 - p_{sn})(SW_{nets} + LW_{nets}) \tag{3}$$





where $SW$ and $LW$ represent the short-wave and long-wave radiative flux components, respectively. Part of the incoming shortwave radiation received by the snowpack is transmitted through the uppermost snow layer, and this energy loss is expressed as $\tau_{n,1} SW_{net\,n}$, where $\tau$ is a dimensionless transmission coefficient, where the snow surface net shortwave radiation is

$$SW_{net\,n} = \alpha_n SW \downarrow \tag{4}$$

where $SW \downarrow$ is the atmospheric downwelling shortwave radiation. The transmission function is described in detail in Decharme

et al. (2016). The total surface net shortwave radiation is defined using a so-called composite albedo defined as

$$\alpha_s = veg\,\alpha_v + (1 - veg)\alpha_g \tag{5}$$

The total surface effective albedo of the snow soil-vegetation composite surface ($\alpha_{eff}$) is then defined by weighting the contribution of each surface:

$$\alpha_{eff} = p_{sn}\alpha_n + (1 - p_{sn})\,\alpha_s \tag{6}$$

with $\alpha_n$, $\alpha_v$ and $\alpha_g$ the snow, vegetation and ground albedos, respectively. Note that no explicit shortwave transmission through the canopy is modeled.

The net longwave radiation for either surface is defined as

$$LW_{net\,X} = \epsilon_X \left( LW \downarrow - \sigma T_{X,1}^4 \right) \tag{7}$$

where $X$ represents either $s$ or $n$, and $\sigma$ is the Stefan-Boltzman constant, $LW \downarrow$ is the downwelling atmospheric radiation and

$\epsilon$ represents the emissivity. The effective emissivity ($\epsilon_{eff}$) of the surface is then defined in a fashion analogous to the effective albedo as

$$\epsilon_{eff} = p_{sn}\epsilon_n + (1 - p_{sn})[veg\epsilon_v + (1 - veg)\epsilon_g] \tag{8}$$

with $\epsilon_n$, $\epsilon_v$ and $\epsilon_g$ representing the snow, vegetation and ground emissivity, respectively. This effective emissivity is then used to compute the effective surface radiative temperature (from the explicitly computed upwelling longwave fluxes from the snow

and composite surfaces) which is required by the longwave radiative scheme when coupled to an atmospheric model.

### 2.1.3 Snow processes

The snowpack model (ISBA-ES) is a multi-layer snow model of intermediate complexity (Boone and Etchevers, 2001; Decharme et al., 2016). The model current uses a default of 12 layers to model the physical processes involved in the snowpack such as solar energy absorption, compaction, snowmelt, water percolation and refreezing of meltwater. The snow albedo is based on a

snow historical variable and considers up to 3 spectral bands. Readers are referred to the aforementioned references for more details.





### 2.1.4 The snow fraction

In the ISBA composite method, the effective fraction of the grid cell covered by snow ($p_{sn}$) is the average between the fraction of snow covering the vegetation and the one covering the ground. It is calculated as:

$$p_{sn} = veg\, p_{snv} + (1 - veg)\, p_{sng} \tag{9}$$

$$p_{snv} = \min\left(1.0, \frac{D}{D + 2z_0}\right) \tag{10}$$

$$p_{sng} = \min\left(1.0, \frac{D}{D_g}\right) \tag{11}$$

where the $p_{snv}$ and $p_{sng}$ values correspond to snow fraction over the vegetation and the ground, respectively, and $D$ is the total snow depth (m). Note that several options for the parameterizations of $p_{snv}$ and $p_{sng}$ exist in SURFEX, however for the current study, Eq.s 10-11 represent the default used with the mult-layer soil and snow schemes. There is no explicit canopy snow reservoir, thus only a masking effect of the vegetation cover is modeled. In order to avoid excessive bare soil evaporation, the default value of the $veg$ parameter is 0.95 for forests (used herein), $z_0$ (m) corresponds to the surface roughness which is calculated as 0.13 times the vegetation height and $D_g$ (m) is a snow depth threshold set to 0.01 m (the default value). As a result, for a forest patch, the maximum value of $p_{sn}$ reaches a maximum of approximately 0.2 for a forest height of 11 m (corresponding to one of the sites in the current study) as shown in Fig. 2. This implies that part of the soil-vegetation composite surface is always in direct contact with the atmosphere regardless of the snow depth, thus the insulating effect of the snowpack is reduced in a forest compared to a bare or low-vegetation covered surface using the composite option. The reasoning for a parameterization resulting in a low $p_{snv}$ value over forests represents a compromise between insulating the soil surface while not burying a forest which would result in an unrealistic coupling with the overlying atmosphere, notably in terms of the total upwelling shortwave radiative flux.

### 2.2 ISBA-MEB : Explicit Vegetation Canopy

The ISBA-MEB option treats up to three fully coupled distinct surface energy budgets (Fig. 1.b) which are: the snow surface, the bulk vegetation canopy and the ground, which is characterized in the current study as a litter layer (Napoly et al., 2017). The reader is referred to Boone et al. (2017) for an extended description of the various assumptions of the MEB approach, its full set of governing equations and its numerical aspects. Compared to the classic ISBA approach, there are two additional prognostic heat storage variables, which are the vegetation temperature, $T_v$ and the litter temperature $T_L$. There are also three new hydrological prognostic variables; the snow liquid water equivalent intercepted by the vegetation canopy, $W_{rn}$ (kg m$^{-2}$ ), and the liquid and liquid water equivalent ice stored in the litter layer, $W_l$ (kg m$^{-2}$ ) $W_{li}$ (kg.m$^{-2}$ ), respectively. Also, note that the vegetation fraction parameter used to aggregate soil and vegetation properties in the composite method, $veg$, is not used in ISBA-MEB since the canopy and soil properties are modeled explicitly.





### 2.2.1 Energy budget

The MEB coupled energy budget equations includes an additional energy budget for the bulk vegetation canopy, and in the current study, the additonal litter energy budget equation is also incldued, which results in a modified upper boundary condition for the uppermost soil temperature. The new and modified energy budget equations are:

$$\mathcal{C}_v \frac{\partial T_v}{\partial t} = R_{net\,v} - H_v - LE_v + L_f \Phi_v \tag{12}$$

$$\mathcal{C}_l \frac{\partial T_l}{\partial t} = (1 - p_{sng})(R_{net\,l} - H_l - LE_l) + p_{sng}(G_{nl} + \tau_{n,N_n} SW_{net\,n}) - G_l + L_f \Phi_l \tag{13}$$

where $T_v$ and $T_l$ are the temperatures (K) of the bulk-vegetation and litter layers, respectively, while $\mathcal{C}_v$ and $\mathcal{C}_l$ correspond to the effective heat capacities (J K$^{-1}$ m$^{-2}$). $R_{net\,v}$, $R_{net\,l}$, $H_v$, $H_l$, $LE_v$ and $LE_l$ (W m$^{-2}$) represent the same quantities as in Eq.s 1-2 but for the bulk vegetation and litter layers. Note that $\mathcal{C}_v$ includes the heat capacities of intercepted solid and liquid water. Note that the snow surface energy budget equation (Eq. 2) is unchanged, however, the definition of the net radiation term has. $G_{nl}$ (W m$^{-2}$) is the conductive heat flux between the lowest snow and the litter layer, and $G_l$ (W m$^{-2}$) is the conductive flux between the litter and the uppermost ground layer. Thus, in contrast to ISBA, the uppermost soil temperature in MEB is only modulated by conductive heat flux divergence and phase changes as

$$\mathcal{C}_{g,1} \frac{\partial T_{g,1}}{\partial t} = G_l - G_{g,1} + L_f \Phi_{g,1} \tag{14}$$

where $\mathcal{C}_{g,1}$ represents the surface soil heat capacity (i.e. with no vegetation effects included). As in the equation of Section. 2.1.1, water phase change terms, $\Phi_v$ and $\Phi_l$, are included for the vegetation and the litter respectively.

### 2.2.2 Radiative transfer through the canopy

MEB represents the explicit radiative transfer through the vegetation for short-wave and long-wave fluxes using classical approaches and it is fully described in section 2.4.2. of Boone et al. (2017). A few key aspects which are pertinent to the current study are described herein. The model uses the classic representation of the canopy as plane parallel surface with a canopy absorption defined as:

$$\sigma_{LW} = 1 - \exp(-\tau_{LW} LAI) \tag{15}$$

where $LAI$ corresponds to the leaf area index (m$^2$ m$^{-2}$) and $\tau_{LW}$ is a coefficient which is set to 0.4 as a default. The model results can be impacted by this parameter, and some sensitivity tests are presented in Section 4.2.1. Note that compared to the composite scheme, MEB increases the downwelling longwave radiation (towards the soil and snowpack) and thus the below-canopy net longwave radiation by including an emission from the canopy (which can be significantly larger than the atmospheric component in cold or dry climates such as in the current study).

The shortwave radiative transfer scheme is described in Carrer et al. (2013). It uses an explicit multi-layer computation that accounts for different characteristics of the vegetation such as the leaf area index, the clumping index, direct and diffuse





radiation components, the thickness of the leaves and the zenith angle. The main outputs are the bulk-canopy reflected, transmitted and absorbed radiation components, and the corresponding photo-synthetically active radiation (PAR) used within the photosynthesis scheme. Compared to the composite scheme, the main impacts are that the downwelling radiation at the surface ground and snowpack is attenuated mainly as a function of the $LAI$. In addition, because the snowpack is generally below the canopy (for the forest heights considered in the current study), the up-welling shortwave radiation is generally significantly reduced compared to the composite scheme since the forest can effectively mask the surface. In the current study, the reflected shortwave radiation is merely a diagnostic, but in a coupled atmospheric model, the shortwave exchange can be significantly modified (the total upwelling shortwave radiation can be significantly reduced in Boreal forest zones: but exploring this impact is beyond the scope of the current study).

### 2.2.3 New and modified snow processes

Both the composite and MEB options are coupled to the identical version of the ES snow scheme. The impact of using the MEB option on snow processes compared to the composite version can be briefly summarized as:

- Only the snow fraction over the ground is considered ($veg = 0$ in MEB) so that the snow fraction is simply defined as $p_{sn} = p_{sng}$. This implies that $p_{sn}$ is generally much closer to unity for MEB than for ISBA in forests (based on Eq.s 9-11 and the discussion in Section 2.1.4). This implies a greater coverage of the ground by snow in MEB, while the canopy is totally exposed to the overlying atmosphere in MEB. Note that when snow depth becomes comparable with the height of the vegetation (for example, for shrubs or grasses), another parameter, described in Boone et al. (2017) is introduced. However, it is not relevant in the current study for the forest heights considered herein.

- In MEB, it is currently assumed that the impact of intercepted snow on the total canopy albedo is negligible. This is based on the results of Pomeroy and Dion (1996). They indicated that the scattering and multiple reflections of light due to intercepted snow, combined with the high probability for the reflected light to reach the underside of an overlying branch and leaves, implies that trees actually behave as light traps. They concluded that intercepted snow has no significant impact on the canopy shortwave albedo or on the net radiative exchange.

- The fluxes from the snowpack are calculated using the specific humidity and temperature of the so called "canopy" air space (Fig. 1.b) instead of the forcing "air" layer when using ISBA (Fig. 1.a). This permits some feedback between the surface and the atmosphere on the fluxes.

- In MEB, the below canopy wind speed which impacts the ground-based snowpack is reduced owing to the attenuation of the wind speed due to vegetation which would tend to reduce sublimation, however, the fractional coverage is generally considerably larger thus generally snowpack sublimation is increased. In addition, sublimation can also occur from intercepted snow.

- An explicit canopy snow reservoir is considered in MEB, which includes interception, unloading, and both freezing of intercepted liquid water and melting of snow.





# 3   Data

The BERMS program (Boreal Ecosystem Research and Monitoring Sites) sites are used in this study. The studied period ranges from 01, January 1999 at 00:00 to 31, December 2010 at 23:30 UTC, corresponding to twelve years of measurements.

The three sites are located in Saskatchewan, Canada and are described in detail in Bartlett et al. (2006). Their distinguishing characteristics are listed up in Tab. 1 can they be briefly summarized as:

- OAS : This site is dominated by 21 m average-height Old ASpen which naturally regenerated after a fire in 1919. A 2 m height under-story composed mainly of hazelnut is present. The ground is characterized by (from the surface downward) an 8-10 cm layer of forest litter, a peat layer, and finally a sandy clay loam soil.

- OBS : The Old Black Spruce is the dominant tree species of this site. Trees have an average height of 12 m. The understorey is comprised of shrubs and herbs, mosses and lichens, situated on sandy loam and sandy soil.

- OJP : The Old Jack Pine site is approximately 14 m high and is composed of a very sparse understorey (alder, bearberry, cranberry and lichens), over a coarse sandy soil.

The full set of meteorological observations needed to force an LSM (downwelling all-wavelength solar and atmospheric
radiation fluxes, air temperature and humidity, pressure, liquid and solid precipitation, and wind speed above the forest canopy) is available at half hourly time steps over the full period, along with data which enable a detailed description of the vegetation characteristics (such as $LAI$, albedo, see Tab. 2).

In order To evaluate the model performance, measurements of turbulent fluxes, upwelling short and longwave radiation, soil temperature and volumetric water content profiles are also available at a 30 minute time step. Snow depth was measured every
30 minutes during the duration of the ground-based snowpack. In addition, manual measurements of snow water equivalent were made up to six times per year. The observed shortwave radiation being transmitted through the canopy (reaching the ground or snowpack surface: $SW_g$) was derived from the Photosynthetically Active Radiation ($PAR$). The data were filtered to account for measurement error due to a direct flux on the sensor around midday, which caused very high peaks. The filter is based on the surrounding three points, using a threshold:

$$\delta = \mathrm{abs}[SW_g[i] - 0.5(SW_g[i-1] + SW_g[i+1])] \tag{16}$$

so that we reject the observation at $i$ if $\delta \geq 100$ W m$^2$. Note that this threshold is somewhat arbitrary because the anomalous peaks are quite large relative to the surrounding values and generally last one time step. Due to the lack of a frost and snow cleaning system on the PAR sensor, we did not use measurements corresponding to these conditions in the evaluation.

The energy balance closure for these sites has been calculated as:

$$closure = \frac{\overline{H + LE}}{\overline{R_{net} - G - S}} \tag{17}$$

where the overbar corresponds to averages over the study period. The storage and ground heat fluxes are not considered here as they were not measured. In addition, it was assumed that they are, on average, negligible compared to the net radiation





when averaged over such a long period. The energy balance closure was 84, 91 and 90% for the OBS, OJP and OAS sites, respectively. This closure is deemed to be satisfactory for the analysis in the current study, especially with respect to the study
of Wilson et al. (2002) which found an average closure of 80% over the Fluxnet network sites.

## 4 Results

For all of the simulations, options for the explicit multi-layer vertical soil heat and water transfer (DIF) and ground-based snowpack (ES) are used, along with the Ag-s stomatal resistance formulation (described in section 2), so that only the impact of the MEB option is evaluated. In the following, we will refer to the different experiments as: 'MEB' for the experiment using
the new Multi Energy Balance option and 'ISBA' for the default experiment. To evaluate the new option, a statistical analysis is performed which is based on simulated fluxes, soil variables and the snowpack characteristics. Then, the study focuses on some specific periods where snow plays a key role governing the surface and sub-surface processes. Finally, a sensitivity analysis is performed to test several new MEB parameters that are the most likely to influence the snow processes. Model input parameters have been chosen which correspond to site measurements where possible. For the remaining parameters, we use
the physiographic database developed for SURFEX (ECOCLIMAP, Champeaux et al.,2005) and the soil parameters from the HWSD data-set (harmonized world soil database, Nachtergaele and Batjes, 2012). The main parameters are given in Tab. 2.

### 4.1 Evaluation

#### 4.1.1 Energy Fluxes

One of the most critical fluxes in coupled land-atmosphere simulations over cold regions is the upwelling shortwave radiation,
$SW \uparrow$. The simulated flux is relatively close between the two experiments and to the observations as shown in Fig. 3b and Fig. 4d,e,f, with averaged RMSE over all sites and years of 7.7 and 7.1 W m$^{-2}$ for MEB and ISBA, respectively (Tab. 3). Improving the modeling of the reflected solar radiation would mostly consists in improving the quality of the input parameters, i.e. albedos (visible and near infrared values for the soil and the vegetation) and $LAI$. At the deciduous OAS site, in winter, the $LAI$ is low (about 1.0 m$^2$ m$^{-2}$) and the $SW \uparrow$ is overestimated, notably in MEB. We suspect that a stem area index ($SAI$)
should be explicitly considered to lower the effect of snow below the canopy on the effective albedo, especially on such a forest of 22 m height, consistent with results from Napoly et al. (2017). The solar radiation that passes through the canopy is only modeled with the new MEB option. When data were available, the simulation of the radiation that is transmitted through the canopy is rather well modeled (Fig. 3,a). Unfortunately, the quality of the data was not sufficient enough when $LAI$ was low at the OAS deciduous site to confirm the assumption of the importance of including a $SAI$. In winter, solar radiation remains
relatively low at this site and barely affects the surface energy balance so that this issue is not addressed for the moment.

The impact of MEB for the OBS site is the opposite to that at the OAS site. At this site, the $LAI$ is relatively large (3.5 m$^2$ m$^{-2}$, see Table 1). Thus, in MEB, the total effective surface albedo is approximately equal to the canopy albedo. In ISBA, there is always a fraction of snow visible to the overlaying atmosphere. Even though it is relatively low (from 10-20% as shown in





Fig. 2), it can result in an overestimation of the total reflected shortwave radiation, especially when the snow is fresh and the

snow albedo is relatively large compared to that of the vegetation. This effect is seen in Fig. 4e. The ISBA bias arises mainly from an over-estimation of the effective surface albedo in February and March (not shown).

Sensible heat flux ($H$) is well simulated over the three sites with MEB compared to ISBA as shown in Fig.s 3d and 4g,h,i, with an average RMSE of 48.4 W m$^{-2}$ (respectively 58.9 W m$^{-2}$) and average BIAS of 4.1 W m$^{-2}$ (respectively -1.0 W m$^{-2}$). This result is consistent with Napoly et al. (2017) who showed that the large overestimation of the ground heat flux diurnal

amplitude from ISBA, confirmed in this study (Fig. 3,f), results in a lack of energy in turbulent fluxes and mostly in $H$ as $LE$ is limited, notably by the evaporative demand. This overestimation is largely decreased with MEB due to the shielding effect of the canopy (thereby reducing the net solar radiation at the below canopy surface) and the insulating effect of the explicit litter layer reduces the heat exchanges with the below-surface soil layers. During periods with snow cover, this improvement is even more marked due to the presence of the snowpack and is investigated in the next section.

Simulations are also improved for the latent heat flux ($LE$) with an average RMSE of 37.1 W m$^{-2}$ for MEB and 47.3 W m$^{-2}$ for ISBA and an average BIAS of 6.6 W m$^{-2}$ for MEB and 9.9 W m$^{-2}$ for ISBA. The main differences appear during spring when ISBA tends to overestimate the total evapotranspiration mainly owing to an excessive soil evaporation (despite the fact that only a 5% soil fraction is prescribed (RMSE and bias are shown in Tab. 3 and Tab. 4, respectively). The default $veg$ value of 0.95 for forests has been tuned to avoid excess bare-soil evaporation in ISBA. In MEB, no tuned parameter is

required to limit baresoil evaporation and it is generally lower than in ISBA owing to a lower surface roughness length (since the surface roughness of the soil is fixed to a few cm at most in MEB, but can be several 10s of cm in ISBA since the soil and vegetation properties are aggregated) and diminished wind speeds owing to the frictional effects of the canopy. In addition, the total annual bare-soil evaporation is further reduced in MEB over the seasonal cycle since the ablation is later. Finally, the litter layer also has an impact on reducing the ground evaporation and further explanation will be given in the next section.

The sublimation of snow represents 27% (12% from the snowpack itself and 15% from the intercepted snow by the canopy) of the total snowfall using MEB whereas it is only 2% with ISBA. This change occurs for essentially two reasons: (i) the snow fraction parameterization gives a low value of snow cover for ISBA compared to MEB (Fig 2) which weights the fluxes, (ii) with MEB, the interception of the snow by the canopy is explicitly considered and allows more sublimation. Even if no observations can confirm these differences, studies have estimated that in forests, sublimation might represent several 10's

of percents of the annual snowfall (Pomeroy and Dion, 1996) and may exceed 30% (Montesi et al., 2004). More recently, Molotch et al. (2007) measured a ratio between sublimation of the snowpack and total sublimation of snow of 0.45 for a forest in Colorado (at 3000 m), which is quite close to the values of 0.42, 0.45 and 0.51 found here for the OJP, OBS and OAS sites, respectively.

### 4.1.2 Snow

The 12-year average annual cycle of the snowpack evolution is shown in Fig.6 for the ISBA (blue curve) and default MEB (red solid line) simulations. The statistical scores calculated on the *last day of snow* when comparing simulation to measurements are shown in Table 5. The *last day of snow* was defined using the two following conditions: (i) the first time when $SND < 0.2$





m was identified, and (ii) the average $SND$ over the ensuing 2 weeks remained below this threshold value. This simple criteria was found to determine the timing of the melt of the snowpack quite accurately without being mistaken with a possible late

snow event. Also, this result is not very sensitive to the chosen threshold value: the idea is simply to eliminate short term snow cover events occurring after the main ablation. The average and standard deviation of the BIAS between modeled and observed *last day of snow* are shown in Tab. 5. With the ISBA option, snow melts on average 24 days too early. Using MEB leads to an improvement in the simulation of the fluxes shown in the previous section as well as the snowpack depth with an average RMSE of 5.1 cm and BIAS of 0.5 cm (Tab. 8). The most important effect appears in springtime when MEB simulates ablation

later with an average BIAS in *last day of snow* of only 1 day (too early) averaged over the 3 sites and full time period. The relatively less frequently-observed values of the snow water equivalent $SWE$ allow a confirmation of the good representation of the timing of snow melt. Also, Fig. 5 seems to indicate that the snow density is well modeled since underestimation or overestimation of $SND$ and $SWE$ are consistent for both models.

In order to better illustrate the ability of the model to represent the snowpack below the canopy, Fig. 5 shows the total snow

depth ($SND$) and $SWE$ evolution over three fairly representative years of the studied period (early 2001 to early 2004). The ISBA average snowpack is simulated with a RMSE of 9.1 cm and BIAS of -1.6 cm (Tab. 8) during this period: the errors mainly arise because the melt of the snowpack occurs too early in the spring season and this behavior is consistent for all years studied. Fig. 7 displays different parameters at the OJP site from 03/25/2004 to 03/31/2004 which correspond to a melting period. The significant overestimation of the ISBA surface soil temperature fluctuations is obvious (Fig. 7,c) as it almost perfectly follows

the temperature measured at 5 meters above the soil. Thus, it leads a large conductive heat flux between the soil and the snowpack (Fig. 7,d) on the order of several hundred W m$^{-2}$, which is unrealistic compared to net radiation (Fig. 7,e). This is explained by the relatively low fraction (10%, see Eq. 9) occupied by the snowpack for a forest in the composite model ISBA. This fraction allows the model to simulate a rather good effective total albedo (Fig. 3,c), but as a result, approximately 90% of soil is not shielded by snow and is strongly coupled to the atmospheric forcing.

As spring begins, the atmospheric temperature gets closer to $0^o$ C and solar radiation starts to increase. With ISBA, the ground temperature can easily rise to over $0^o$ C as the heat capacity of that layer is low and part of the ground surface is directly exposed to the atmosphere (again, owing to a relatively low $p_{sn}$ value compared to MEB). Once the ground temperature exceeds $0^o$ C, the conductive flux between the snow and the ground (Fig. 7,d) is negative, indicating that the ground is warming up the snowpack from below. The early melt of the snowpack in ISBA is thus due, in large part, to that energy received from

the combined ground-vegetation layer (Fig. 7,c). In MEB, the insulation of the soil from the snowpack is total as the horizontal coverage of the snow is more realistic. The flux coming from the ground is very close to 0 W m$^{-2}$ (Fig. 7,d). Thus, the melt of the snowpack comes almost entirely from above (as the snow becomes thin, some solar energy can warm the ground below the snowpack thereby melting the snow from below using MEB also, but this effect tends to be quite small compared to melting induced by surface flux of heat into the snowpack). The net radiation (Fig. 7,e) received by the first layer of the snowpack is

higher in MEB than in ISBA due to the longwave enhancement effect, and this causes the snowpack to melt at a speed more comparable to the measurements (Fig. 7,a).





A period before the ablation of the snowpack is shown in Fig. 8. Certain fluxes from the ISBA energy budget ($H$ and $G$) are quite different compared to observations. Indeed, because the snowpack does not cover the full grid, the available energy is used to warm up or cool down the surface soil temperature which provokes strong amplitude of $G$ instead of being released to
the atmosphere through $H$. With MEB, two prognostic temperatures (Fig. 1,b) are used, $T_l$ for the surface litter, which barely varies in time and $T_v$ for the exchanges with the atmosphere which is related to an explicit heat capacity of the vegetation (lower than the composite heat capacity of ISBA). These two temperatures, which are totally uncorrelated due to the snowpack which occupies the full surface of the ground, lead to a much improved modeling of energy fluxes.

### 4.1.3   Soil Temperature and Water Content

The overestimation of the ground heat flux amplitude by ISBA not only impacts energy exchanges with the atmosphere through $H$ but also the soil temperatures. With the direct contact of about 90% of the composite layer with the atmosphere, the soil temperature at a depth of 10 cm calculated from ISBA can drop to below -20°C in winter months (Fig. 9) whereas observed temperatures at this depth are only slightly negative: this feature is common for the three sites and during the entire period (results are shown here for 3 years for ease of visual inspection). This leads to a significant temperature BIAS averaged over
the three sites and full time period at 10 cm depth of -2.9 K and an RMSE of 6.8 K. Owing to the insulating effect of the snowpack, MEB is much closer to observations with an average BIAS of 0.1 K and RMSE of 2.0 K.

In ISBA, the increased exposure to cold atmospheric conditions leads to a cold bias which extends to at least 1 m depth throughout the season (the 12-year average seasonal cycle of soil temperature for the three sites is shown in Fig. 10,b), which is the maximum depth of the soil temperature observations. This leads to significantly more soil freezing with depth during early
winter. In spring, even after the snowpack ablation, the frozen water component remains significant in the deep soil layers. This cold bias is mainly owing to the under-estimated impact of the insulating effect of snow (a low $p_{sn}$ in ISBA) Finally, note that even if the MEB-simulated soil temperatures warm a bit more slowly than the observations (as evidenced by the small delay and slight tilt of the annual temperature wave compared to the observations indicating a bit more inertia in MEB verses the observations), MEB provides a much improved soil temperature simulation.

The near surface (7.5 cm depth) modeled soil liquid water content (Fig. 9) agrees reasonably well with the observations for both versions of the model except for the OJP site. The overestimated values at this site are likely to be due to the definition of the soil characteristics which are defined based on soil texture information. A noticeable difference between the two models is that the water content curves are generally more flat with the MEB option in months outside of summer than ISBA which is in better agreement observations. Indeed, ISBA occasionally melts the entire snowpack erroneously as shown in Fig. 5, leading
to short periods of ice melting and unrealistic peaks of liquid water content. The impact of changes in soil freezing between MEB and ISBA on drainage and runoff are thus expected in regional or global studies and this could further have an impact on the hydrological cycle (notably river flow) in such regions. This issue using ISBA was identified by Decharme et al. (2019), who found a precocious springtime peak of river discharge over all Arctic basins. Thus, it is anticipated that MEB should at least improve this bias in future large-scale hydrological studies using SURFEX in both coupled and offline modes.





## 4.2 Sensitivity tests


Several sensitivity tests were performed and the results are summarized here. The analysis focuses on three parameters and one process for which the values are considered to be somewhat uncertain and for which the snowpack is potentially sensitive. For each parameter, values were tested for each site over a range (either based on the literature or physical reasoning) and compared to the default value defined in Boone et al. (2017). Statistical scores were calculated only when snow was observed 440 on the ground.

### 4.2.1 Canopy longwave radiation transmission

The $\tau_{LW}$ parameter is an absorption coefficient which is used to calculate the $LW$ radiation transmitted through the canopy (Eq. 15), and it weights the canopy emission to the soil and the atmosphere. The original default value of $\tau_{LW}$ is 0.5 (Boone et al., 2017) and values from 0.1 to 1.0 using increments of 0.1 are tested in the current study. This range covers values based on 445 a literature survey as discussed in the aforementioned reference, although 1.0 is quite large and is tested simply for numerical reasons. As $\tau_{LW}$ increases, the canopy transmission decreases and the canopy emission increases (increasing the longwave radiation received by the snowpack). Fig. 11 shows the RMSE calculated for each value of this parameter over the 12-year period for each site for the identified 4 most impacted state variables and flux. The sensitivity to this parameter is relatively high, notably for low values. For each variable, errors are lowest for $\tau_{LW}$ values in the range 0.3-0.4. For further increases in 450 this parameter, the RMSE stabilizes for $LW \uparrow$ and $H$, and starts to increase for $SND$ and $T_G$. This behavior is consistent for all three sites (as shown). Thus, the value of 0.4, quite close to the default one, and it has been selected to be the new default value (and it has been used for the results presented in previous sections).

### 4.2.2 Litter Thickness

The litter thickness has been identified as a key parameter of MEB (Napoly et al., 2017). Indeed, it affects both the thermal and 455 hydrological fluxes and state variables in the model since it's thickness modules the litter surface energy budget and it's water storage capacity. Its value can be very specific for a particular site and can evolve in time, and, in addition, values are hard to determine at a large scales. Its variation is generally in the range from 0.01 m to 0.10 m based on a literature survey shown in Napoly et al. (2017). In addition, this range has been selected for essentially two additional reasons: the first is that model tests have shown that results degrade for thicknesses below 0.01 m since the assumption of the existence of a continuous litter 460 layer becomes physically dubious and if it is too thin, not to mention numerical issues can arise since the surface energy budget is computed within this layer. Second, when the layer exceeds approximately 0.10 m, the diurnal cycle is highly damped (to levels which are unrealistic): a multi-layer litter model would be preferable within the MEB model structure in this case. These two issues are discussed in more detail in Napoly et al. (2017): based on the aforementioned study, the MEB default value for litter thickness is constant in time and set to 0.03 m. However, in the current study, tests showed no significant sensitivity to this 465 parameter during snow periods (<10% variation of RMSE of the tested values compared to the default value) on most of the state variables and total surface-atmosphere fluxes. Only the soil temperatures were found to be significantly impacted, with





optimized RMSE values of 1-2 K obtained with litter thickness values at or above 0.06 m, instead of 3-4 K with 0.01 m. Note that these differences are essentially due to the initial state of the ground when the snow season begins since the litter effect is active during the entire year.

### 4.2.3 Roughness length for heat and water vapor

The ratio of the vegetation roughness lengths of momentum to heat (and vapor), $r_{z0} = z_{0,v}/z_{0,vh}$, which is a SURFEX input parameter which is used to diagnose the roughness for heat and vapor fluxes from the prescribed momentum roughness length, $z_{0,v}$, is tested. The lower $r_{z0}$ is, the higher the turbulent fluxes become: in ISBA, the default value is $r_{z0} = 10$ while it is $r_{z0} = \exp(1)$ in MEB (Napoly et al., 2017) following Lo (1995) and Yang and Friedl (2003) who propose values more adapted for forest covers. The uncertainty associated with this parameter motivated this sensitivity test, and values from 1 to 10 were tested. Note that for certain local scale studies with ISBA, values in excess of 10 have been used, however, for current applications in hydrology and atmospheric modeling in SURFEX, the default of 10 is used so this is the limit used herein. As it turns out, the sensitivity to this parameter during the snow period is very low for the three studied sites with a maximum variation of 3 W in the RMSE in $H$ and $LW \uparrow$ compared to the results obtained using the default values, therefore in conclusion no modification was made.

### 4.2.4 Snow interception

The snow interception parameterization in MEB is based on Hedstrom and Pomeroy (1998) and the implementation in MEB is described in detail in Boone et al. (2017). To investigate the sensitivity of the interception of snow by the canopy on the snowpack, the simulations were repeated using MEB with the maximum interception storage set to zero, thereby effectively turning "off" the snow interception and loading parameterization. It was found that for these particular sites , the process of snow interception by the canopy vegetation has only a mild impact on the snowpack below the canopy and a fairly small impact on the fluxes to the atmosphere. The impact of removing the snow interception on simulated 12-year average annual cycle of snow depth is shown in Fig.6 in which the MEB simulation without this process is represented by the red-dashed curve. The RMSE of the simulated snow depth varies by less than 10% between this test and the default MEB simulation averaged over the 3 sites. The maximum snow peak is increased in average by 10% (3 cm) and the total $LE$ is decreased by approximately 4% on average for all three sites resulting in small compensating increases in $H$. Perhaps most significantly, there is virtually no impact on the $last\ day\ of\ snow$ score defined in section 4.1.2 which corresponds to the main difference (and improvement) when comparing MEB to ISBA. We conclude that even if snow interception is a key physical process in land surface schemes such as MEB (Rutter et al., 2009), it does not have much of an impact for the Berms sites in terms of the improvement of the snowpack modeling, although it does have a relatively large impact on the sublimation, but no specific observations of this flux are available for the sites herein. Therefore, we leave the parameterization with its default parameter values pending the results of future studies.





## 5   Conclusions

The impact of snow conditions on the surface fluxes and state variables simulated using the multi energy balance (MEB)
option, which has recently implemented in the ISBA LSM on the SURFEX platform, is evaluated in this study. The default
representation of the surface energy balance in ISBA consists of a single composite soil-vegetation layer for which physical
parameters for the two surfaces are weighted by a fraction of surface covered by vegetation. The new option improves the
representation of forests through the addition of two explicit layers: a bulk vegetation canopy and a forest surface litter layer. A
new energy budget is computed for the bulk canopy, while the below canopy energy budget is computed for the litter layer. The
evaluation has been carried out using twelve years of observations available from the three Berms (Boreal Ecosystem Research
Study) experimental sites which have been used in numerous studies (e.g. Bartlett et al., 2006) and the recent ESM-SnowMIP
intercomparison study (Krinner et al., 2018; Menard et al., 2020) and can be considered as a benchmark for evaluating LSMs
simulating cold season processes for forested areas.

During periods without snow-cover, comparable results and conclusions from a previous study (Napoly et al., 2017) which
compared ISBA with ISBA-MEB were confirmed. They can be summarized as: due to the shading effect of the canopy layer
and the low thermal diffusivity of the litter layer, the ground heat flux daily amplitude is significantly reduced as well as soil
temperatures daily amplitudes. The result is that the ground heat flux was found to be in much better agreement with the
measurements (RMSE of 47.1 W m$^{-2}$ with ISBA verses 10.9 W m$^{-2}$ with MEB) over the entire 12-year integration period.
The reduced energy used for conductive heat flux (note that net radiation is barely impacted between the two versions) tends to
be manifested as concomitant increases in the daily peak sensible heat flux (RMSE of 58.9 W m$^{-2}$ with ISBA and 48.4 W m$^{-2}$
with MEB) as the latent heat flux is limited by the evaporative demand. In spring, the latent heat flux is also improved mainly
owing to more a limited contribution of the ground evaporation due to the addition of a litter layer (main effect), a decreased
surface roughness, and lower wind speeds compared to ISBA. Available measurements of short wave radiation below the
canopy also show the ability of MEB to model the radiative transfer through the canopy.

During snow periods, MEB provides an improved realism of the decoupling between the atmosphere and the ground below
the snowpack. Since MEB has eliminated the fractional burying of the ground-based snowpack by the vegetation layer (the
$p_{snv}$ parameterization), the below canopy surface relies on $p_{sng}$ uniquely and therefore the ground can be completely covered
and insulated for a relatively shallow total snow accumulation (the default value is 0.10m, which was used in the current study).
With the default composite version of ISBA, this decoupling cannot be represented as an effective snow fraction is calculated
in the range of 10 to 20 %. Consequently, a large fraction of the surface is directly connected with the atmosphere and ground
heat flux becomes large (negative directed downward) when the atmospheric temperature decreases. This leads to a strong
unrealistic cooling of the soil with an average bias for the all sites, depths and years of -5°C compared to -0.1°C with MEB. In
addition to the insulating effect of the litter, this improved representation of the ground heat flux provides energy to turbulent
fluxes (and, again, mostly the sensible heat flux) which were underestimated. The average RMSE for the sensible heat flux
calculated for snow periods drops from 54.9 to 45.0 W m$^{-2}$.





The improved soil temperature simulation (reduced cold bias) was also found throughout the soil column, and was confirmed by the observations which extended to a 1 m soil depth. It should be noted that this significantly impacts soil phase changes in the model and thus, for example, the modeling of the permafrost in forest regions within large-scale coupled-atmospheric or hydrological simulations. This aspect is the subject of current research and and is beyond the scope of this study, but here
we note that ISBA coupled to a river routing scheme tends to simulate a river discharge peak owing to springtime thaw and snow melt in historical offline simulations north of $50^o$ N too early in comparison to observations and this is attributed to a precocious snow melt and soil thaw in Boreal forest regions Decharme et al. (2019).

The impact of the explicit vegetation canopy and litter layer on the snowpack simulation is significant. In general, the snow depth is improved with MEB. The average RMSE calculated for all sites and years is 5.1 cm for MEB while with ISBA it is
9.1 cm. This is due to a general better agreement during the entire season but, in particular, from a better representation during the melting period. Indeed, the snow melts, on average, 24 days too early with ISBA, while the melt occurs only one day early with MEB. This mainly arises in ISBA due to the more direct coupling of the ground with the atmosphere. When the air temperature increases above freezing, the composite layer temperature also warms thereby heating the snowpack from below and provoking melt. In MEB, the snowpack occupies the whole fraction of ground so that it can only melt from its surface
due to a positive energy balance excess. In addition, because of the lower fractional coverage of snow in ISBA, sublimation represents only 2 % of the total snowfall loss, while it represents approximately 27 % when using MEB. About half of this quantity corresponds to snow intercepted by the canopy, the other half is directly sublimated from the snowpack. While there are no direct estimations of sublimation available at the Berms sites, it is found that these values correspond well with the total sublimation and partitioning values quoted in the literature for forested sites.

Two hydrological impacts are to be expected with the MEB option. First, the soil does not freeze as deep or as long. With ISBA, the deep soil can be frozen to well below 1 m depth more than half of the year, while the depth of the 0 C isotherm and the soil water frozen fraction are both considerably less with MEB. At a soil depth of 1 m, on average over all sites and years, the daily temperature falls bellow $0°$ C 33 days in the observations, while it is simulated as 35 days per year with MEB compared to 188 days with ISBA. This effect tends to cause ISBA to have a later peak in total runoff. It will be studied in
global runs coupled with a hydrological model in the near future.

It should be noted that when doing local scale uncoupled studies with ISBA, the default $p_{snv}$ parameterization can be changed such that it rapidly reaches unity after a few cm of snow-cover has developed. The result is a simulation which is more consistent with MEB in terms of soil temperature and snowpack duration. However, such a configuration can not be used in coupled (with an atmospheric model) mode since there would be a huge positive bias in the upwelling shortwave radiation,
notably in Spring, thereby potentially having an impact on the simulated high latitude radiative feedback (which is important to simulate correctly for climate change prediction). In contrast, if the standard $p_{snv}$ parameterization is used in ISBA (similar to the coupled configuration), ISBA tends to produce the effects cited herein. There, MEB has removed this inconsistency.

For application of MEB in spatially distributed applications, such as for NWP or hydrological forecasting, such parameters are referred to as "primary parameters" are generally fixed or prescribed from look-up tables based upon land-use classification
or plant functional types (set in the SURFEX physiographic database ECOCLIMAP): thus care must be taken to define values





and explore model sensitivity. This is in contrast to so-called "secondary parameters" which are derived based upon input primary parameters or input physiographic data (such as $LAI$, soil texture, etc.). As it turns out, the model showed significant sensitivity to only one parameter among the 3 tested, which was the long-wave radiation transmission coefficient. In this study, a slightly lower value of 0.4 was finally chosen compared to the default value of 0.5 from (Boone et al., 2017). The default

values of the other parameters from the aforementioned study were unchanged as a result of this study.

Work is currently underway to use MEB within ISBA for forest covers in many of the applications using SURFEX. For regional (covering France) high resolution (8 km grid) hydrological and surface state forecasting and analysis, Météo-France uses the SURFEX-ISBA-MODCOU hydrometeorological model version 2 (SIM2: P. Le Moigne et al., 2020) and MEB is being tested for future implementation. There are also preparations underway to use MEB in the coupled SURFEX-CTRIP

system in both offline mode and coupled to CNRM-CM (Decharme et al., 2019). Work is also underway to use MEB coupled to the detailed snow process model CROCUS (Vionnet et al., 2012), which is used for, among many applications and fundamental research, operational avalanche prediction for French mountain areas. There are longer term plans to use MEB in the operational regional and global meteorological prediction models AROME and ARPEGE, respectively. Thus, we will continue to evaluate MEB from the local scale (in order to study processes in detail), up to global scales in both offline and coupled

land-atmosphere-hydrology model platforms.

*Code availability.* The MEB code is a part of the ISBA LSM and is available as open source via the surface modeling platform SURFEX, which can be downloaded at http://www.cnrm-game-meteo.fr/surfex/. The developments presented in this paper are available starting with SURFEX version 8.1.

*Author contributions.* AB and AN have contributed to the development and improvement of the MEB code. TW and AN performed the
evaluation step by performing the simulation and comparing the model results with the experimental data. All authors contributed to write the text.

*Competing interests.* No competing interests are present in this study



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





**Table 1.** *Main physical characteristics of the sites.*

| Site | OBS | OJP | OAS |
|---|---|---|---|
| Forest type | Black Spruce | Jack Pine | Aspen |
| Location (°) | 53.99N, 105.12W | 53.92N, 104.69W | 53.63N, 106.2W |
| Canopy height (m) | 11 | 13 | 21 |
| LAI ($m^2.m^{-2}$) | T3.5-3.8 | 2.5-2.6 | winter = 1 & summer = 3.7-5.2 |
| Snow Free Albedo (-) | 0.08 | 0.11 | 0.14 |

**Table 2.** *Model characteristics prescribed for the three different sites.*

| Site | OBS | OJP | OAS | Source |
|---|---|---|---|---|
| vegtype type | 5 : BNE | 5 : BNE | 16 : BBCDS | ECOCLIMAP |
| Vegetation fraction (ISBA only) | 0.95 | 0.95 | 0.95 | ECOCLIMAP |
| LAI ($m^2.m^{-2}$) | 3.65 | 2.55 | 1.0-4.9 | Measurements |
| Vegetation albedo, NIR (-) | 0.12 | 0.18 | 0.26 | Measurements |
| Vegetation albedo, VIS (-) | 0.04 | 0.04 | 0.06 | Measurements |
| Soil albedo, NIR (-) | 0.17 | 0.17 | 0.17 | ECOCLIMAP |
| Soil albedo, VIS (-) | 0.07 | 0.07 | 0.07 | ECOCLIMAP |
| Root depth (m) | 1 | 1 | 1 | ECOCLIMAP |
| Ground depth (m) | 2 | 2 | 2 | ECOCLIMAP |
| Elevation (m) | 629 | 579 | 600 | Measurements |
| Temperature / Humidity measurement height (m) | 25 | 28 | 27 | - |
| Wind measurement height (m) | 26 | 29 | 38 | - |
| Sand (%) | 0.58 | 0.92 | 0.58 | Measurements |
| Clay (%) | 0.1 | 0.03 | 0.27 | Measurements |
| Soil Organic Carbon TOP (%) (0 - 30 cm) ($kg.m^{-2}$) | 18.81 | 18.75 | 21.55 | HSWD |
| Soil Organic Carbon SUB (%) (30 - 70 cm) ($kg.m^{-2}$) | 44.30 | 44.42 | 52.18 | HSWD |





**Table 3.** *RMSE for the ISBA-MEB and ISBA experiments for fluxes SWUP, LWUP, H, LE and G calculated over half hourly data.*

| RMSE (W m$^{-2}$) (MEB / ISBA) | OBS | OJP | OAS | Period |
|---|---|---|---|---|
| SW ↑ | 5.6 / 5.8 | 5.7 / 6.4 | 11.8 / 9.0 | Full Period |
| | 6.1 / 6.9 | 6.2 / 6.5 | 12.5 / 8.1 | Snow Period |
| LW ↑ | 6.1 / 6.7 | 5.5 / 5.6 | 7.4 / 5.6 | Full Period |
| | 6.7 / 7.0 | 5.2 / 6.1 | 7.4 / 5.2 | Snow Period |
| H | 47.1 / 57.0 | 49.4 / 65.9 | 48.7 / 53.7 | Full Period |
| | 43.1 / 53.7 | 46.2 / 60.1 | 46.1 / 50.8 | Snow Period |
| LE | 35.9 / 48.8 | 37.5 / 48.4 | 37.8 / 44.7 | Full Period |
| | 25.9 / 34.2 | 24.5 / 30.1 | 33.0 / 38.7 | Snow Period |
| G | No data | 10.9 / 47.1 | No data | Full Period |
| | | 5.9 / 50.1 | | Snow Period |





**Table 4.** *BIAS for the ISBA-MEB and ISBA experiments for fluxes SWUP, LWUP, H, LE and G calculated over half hourly data..*

| BIAS (W m$^{-2}$) (MEB / ISBA) | OBS | OJP | OAS | Period |
|---|---|---|---|---|
| $SW \uparrow$ | -2.1 / 0.1 | -1.4 / 0.7 | 2.8 / 1.7 | Full Period |
|  | -2.3 / 0.4 | -2.0 / 0.6 | 3.5 / 0.8 | Snow Period |
| $LW \uparrow$ | 0.5 / 0.7 | 0.8 / 1.3 | -2.1 / -0.4 | Full Period |
|  | 0.2 / 1.3 | 1.0 / 2.6 | -2.8 / 0.1 | Snow Period |
| $H$ | 7.9 / 0.4 | -1.6 / -6.0 | 6.0 / 2.6 | Full Period |
|  | 6.8 / 4.6 | 1.9 / 4.5 | 5.0 / 5.1 | Snow Period |
| $LE$ | 7.8 / 12.4 | 8.4 / 11.4 | 3.6 / 5.8 | Full Period |
|  | 4.0 / 4.6 | 3.6 / 3.2 | 2.9 / 3.5 | Snow Period |
| $G$ | No data | 0.2 / 1.1 | No data | Full Period |
|  |  | -0.2 / -3.6 |  | Snow Period |

**Table 5.** *average and standard deviation of the BIAS between model and observations of the last day of snow expressed in number of days. The OAS site has only six years of snow observations compared to nine for both the OBS and OJP*

|  | OBS | OJP | OAS |
|---|---|---|---|
| MEB | $-1.5 \pm 3.8$ | $4.5 \pm 5.5$ | $-6.2 \pm 11.4$ |
| ISBA | $-25.0 \pm 12.1$ | $-20.7 \pm 7.4$ | $-26.7 \pm 4.0$ |




**Table 6.** *RMSE for the ISBA-MEB and ISBA experiments for soil temperature at 5, 20 and 100 cm calculated over half hourly data.*

| RMSE (K) (MEB / ISBA) | OBS | OJP | OAS | Period |
|---|---|---|---|---|
| 5 cm | 3.3 / 9.2 | 2.0 / 7.7 | 2.2 / 8.2 | Full Period |
| | 2.1 / 11.0 | 1.9 / 9.8 | 2.3 / 9.5 | Snow Period |
| 20 cm | 3.3 / 8.1 | 1.5 / 6.3 | 1.8 / 7.0 | Full Period |
| | 2.2 / 10.1 | 1.7 / 8.3 | 1.9 / 8.2 | Snow Period |
| 100 cm | 1.4 / 4.3 | 1.1 / 5.1 | 1.0 / 5.0 | Full Period |
| | 0.7 / 5.1 | 0.9 / 4.8 | 0.9 / 4.9 | Snow Period |

**Table 7.** *BIAS for the ISBA-MEB and ISBA experiments for soil temperature at 5, 20 and 100 cm calculated over half hourly data.*

| BIAS (K) (MEB / ISBA) | OBS | OJP | OAS | Period |
|---|---|---|---|---|
| 5 cm | 1.1 / -2.4 | 0.3 / -3.4 | -0.5 / -3.8 | Full Period |
| | -0.1 / -6.8 | 0.7 / -6.2 | -0.9 / -5.5 | Snow Period |
| 20 cm | 1.0 / -2.9 | 0.2 / -3.8 | -0.6 / -4.1 | Full Period |
| | -0.4 / -6.9 | 0.6 / -6.0 | -0.9 / -5.5 | Snow Period |
| 100 cm | 0.7 / 3.8 | -0.2 / -4.6 | -0.7 / -4.7 | Full Period |
| | 0.1 / -4.5 | 0.5 / -4.3 | -0.7 / -4.6 | Snow Period |

**Table 8.** *RMSE and BIAS for the ISBA-MEB and ISBA experiments for snow depth calculated over half hourly data.*

| (MEB / ISBA) | OBS | OJP | OAS |
|---|---|---|---|
| RMSE (cm) | 3.7 / 8.6 | 4.5 / 8.3 | 7.2 / 10.4 |
| BIAS (cm) | 0.0 / -2.0 | 1.5 / -0.9 | 0.1 / -2.0 |



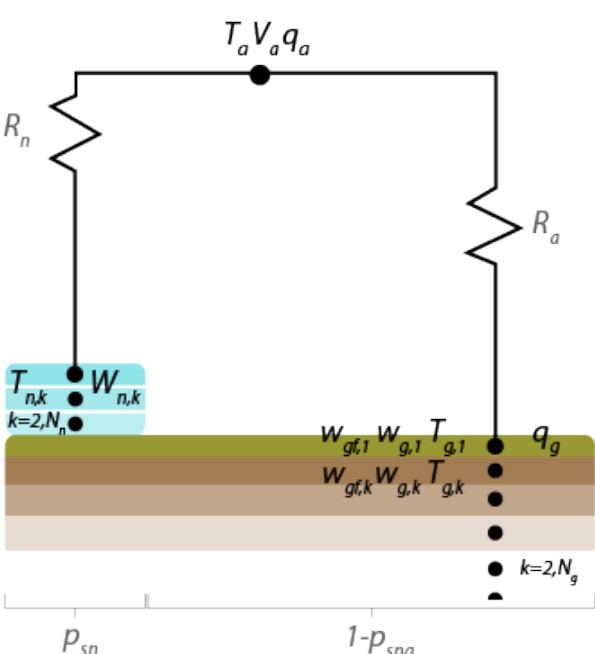

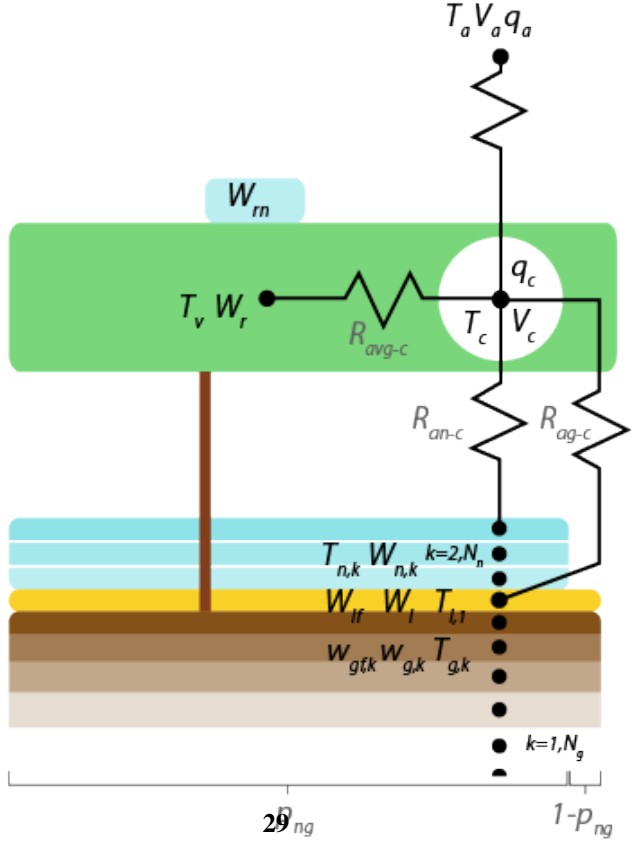





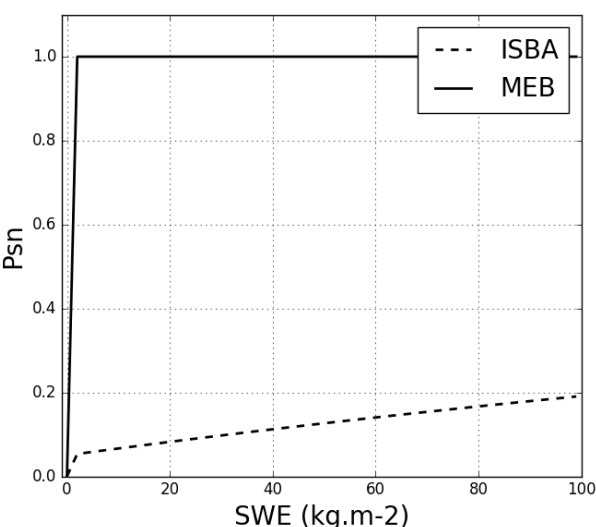

**Figure 2.** Snow fraction for different values of $SWE$ for ISBA and MEB over a forest of 11 meters height.



**Figure 3.** Composite of monthly diurnal cycle at OJP site. MEB is in red, ISBA in blue, measurements are indicated by a solid line and adjusted measurements are represented using a dashed black line. Adjusted measurements respect the energy balance closure and are calculated following the method of Twine et al. (2000). As a visual aid, the area between the latter two is shaded and model outputs should fall within this area.





**Figure 4.** Scatter plots for the different fluxes and the three sites using only observations with snow on the ground. MEB is in red, ISBA in blue.

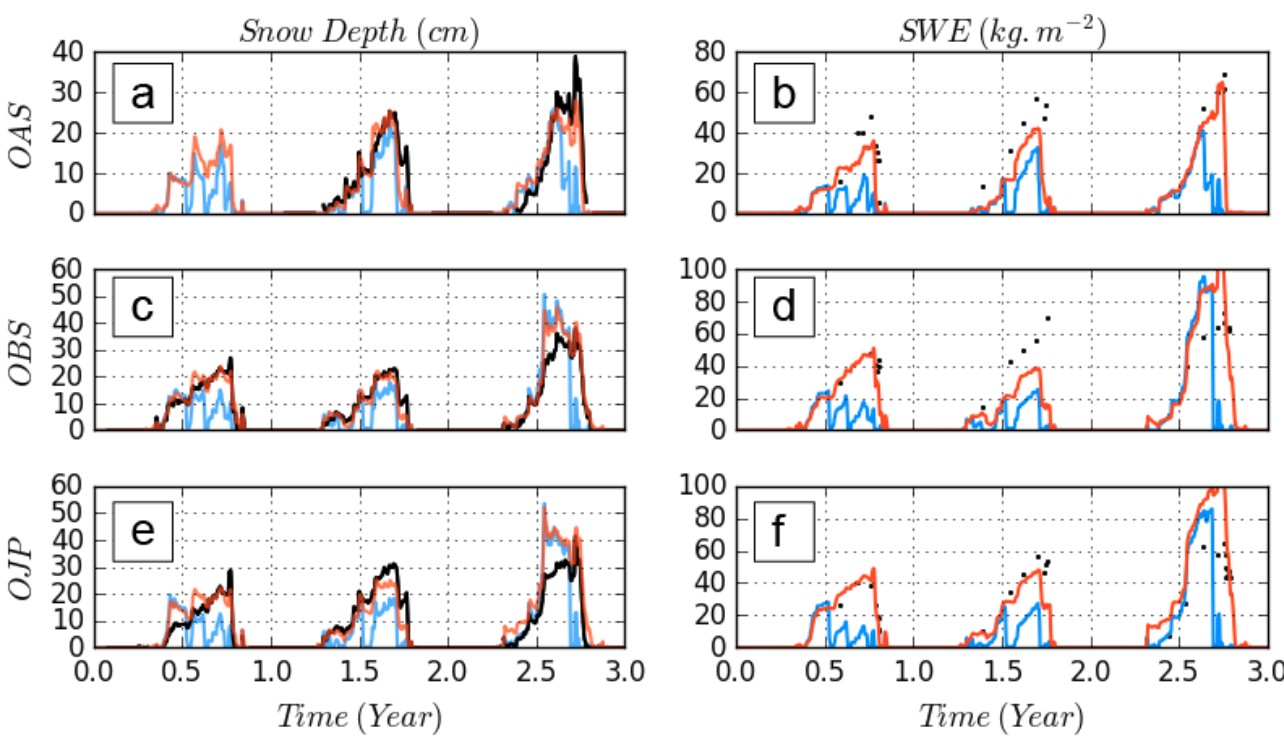

**Figure 5.** Snow depth (left column) and snow water equivalent (right column) for the three sites from 07/01/2001 to 07/01/2004. MEB is in red, ISBA in blue and observations in black.





**Figure 6.** Composite of snow depth annual (July to June) cycles for the three sites from. MEB is in red, ISBA in blue and observations in black. The red dashed curve corresponds to a version of MEB with no snow interception by the canopy.



**Figure 7.** Multiple variables at the OJP site from 03/25/2004 to 03/31/2004 which corresponds to a melting period of the snowpack. MEB is in red, ISBA in blue and observations in black. $T_{G1}$ is the temperature of the first layer of the surface (i.e. the composite for ISBA and the first layer of the ground for MEB).



**Figure 8.** Multiple variables at the OJP site from 01/25/2008 to 02/01/2008. MEB is in red, ISBA in blue and observations in black. For $T$, the red curve is $T_C$ (MEB), the dotted red curve is $T_{G1}$ (MEB) and the blue curve is $T_{G1}$ (ISBA)



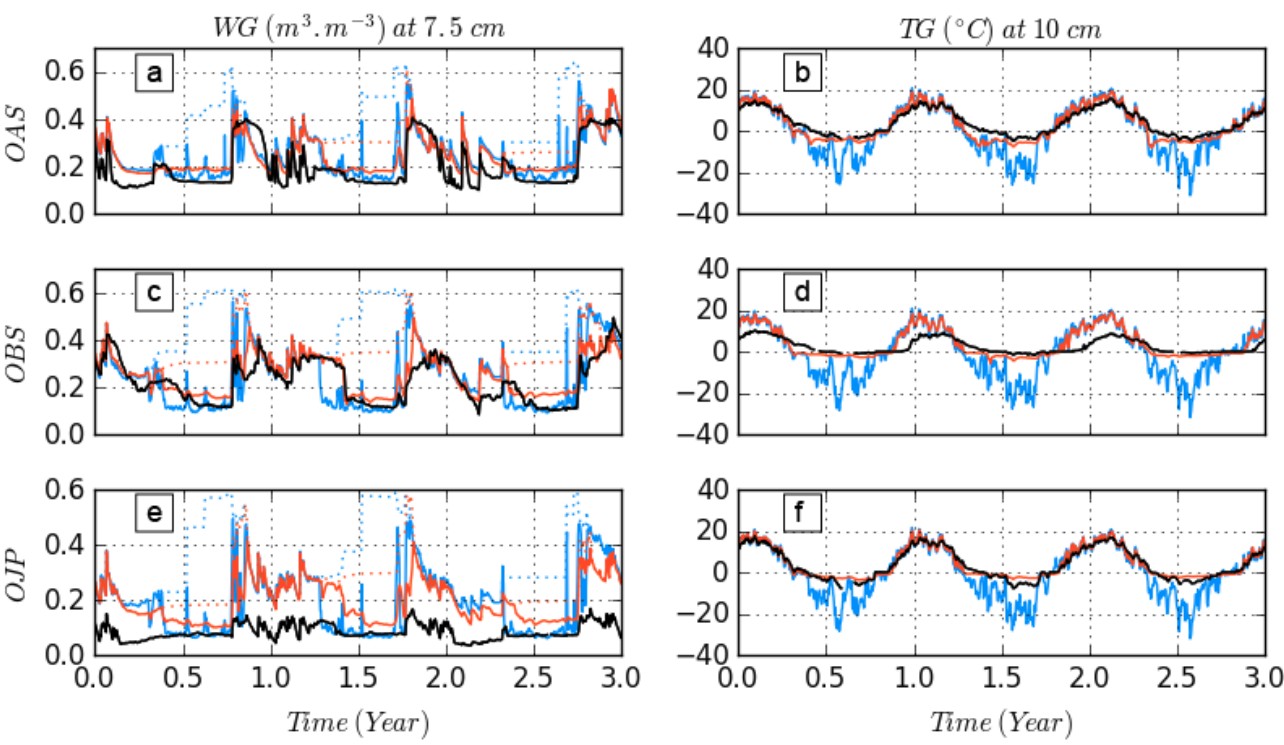

**Figure 9.** Soil water content and temperature at 7.5 and 10 cm deep respectively for the three sites from 07/02/2001 to 07/02/2004. MEB is in red, ISBA in blue and observations in black. On the $WG$ graphs, the dotted lines represent the liquid and solid water.



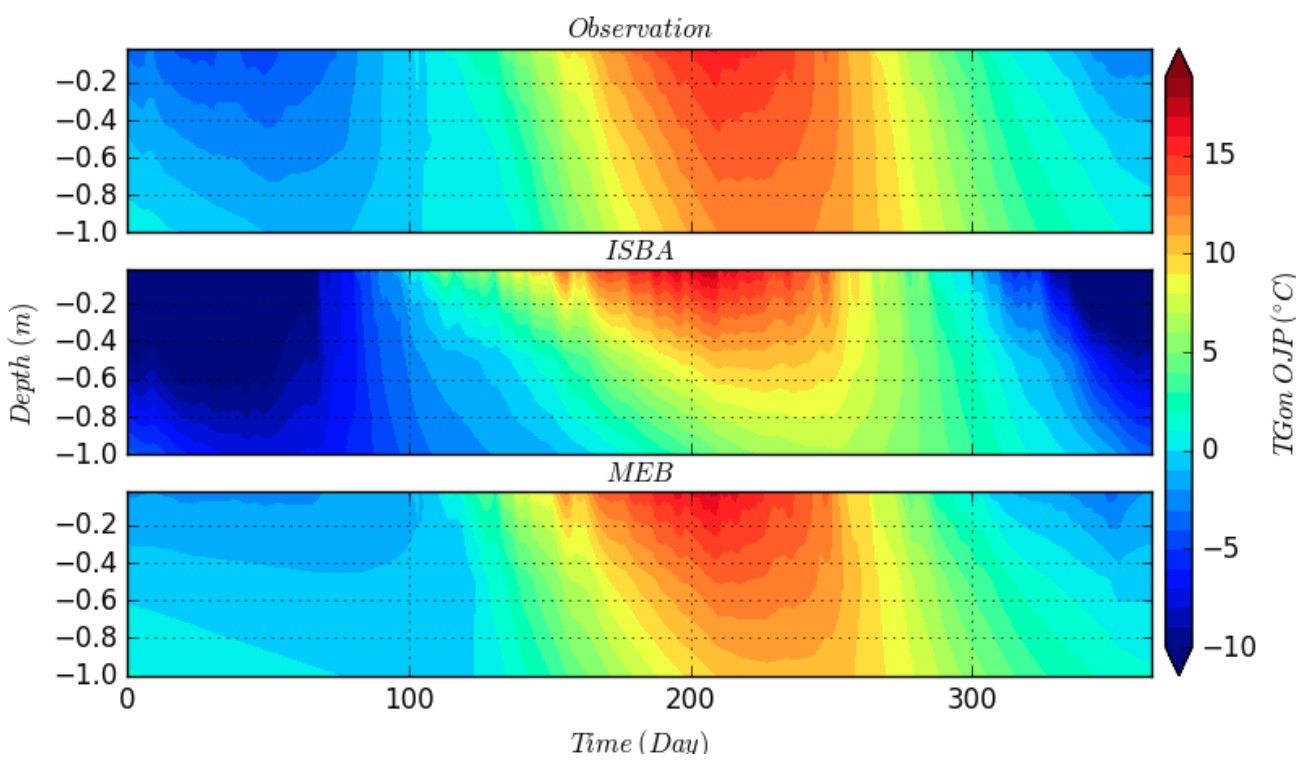

**Figure 10.** Average annual contours of soil temperature between surface and 100 cm deep at the OJP site.



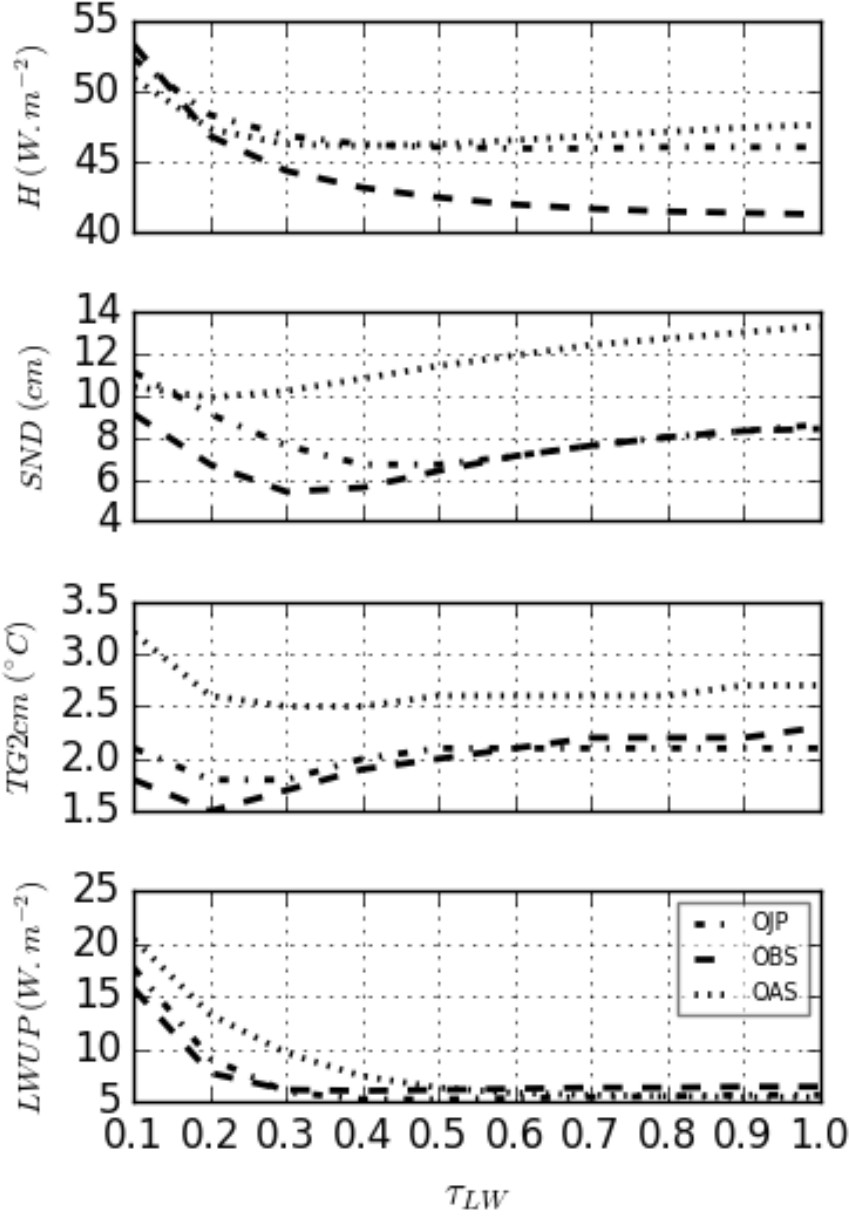

**Figure 11.** RMSE calculated for different values of $\tau_{LW}$ for each site during the snow period and for the sensible heat flux (a), snow depth (b), soil temperature at 2 cm (c) and $LW \uparrow$ (d).