# Peer review of "ISBA-MEB (SURFEX v8.1): model snow evaluation for local-scale forest sites"

_Geoscientific Model Development, 2020_

## Referee Comment (RC1) · Anonymous Referee #1 · 2 Sep 2020

General comments:

The manuscript presents valuable model developments focused on forested snow-process modeling in a popular land surface model. The model developments are structural in nature and specify the relationship between vegetation and the snow surface simulated below the vegetation canopy. As compared to the undeveloped land surface model, simulated snow duration and soil temperature are significantly improved. The developments presented here are valuable to the scientific community and deserve prompt publication pending a small number of minor and technical revisions. These revisions are mainly contextual and should not change any major findings or results.

Scientific comments:

I really enjoyed this manuscript and commend the authors for their hard work. I have three overarching comments:

1. I liked the organization of this manuscript. The results were presented in a fashion that made both the model developments and findings easy to understand. On my second readthrough, I found that my mind was already primed to identify the issues with the default ISBA model that made modeling snow in forests troublesome. However, when I first read through this manuscript, I wanted to know more about the sources of model errors before getting into the model details. Only in lines 195 – 200, and the following sections describing MEB, did I start to understand what was being corrected. Specifics about the changes to the model framework and how that influenced the snowpack could be put earlier to prime the reader for what to expect. This need not be lengthy (only a few sentences) and could be included in an individual paragraph, headed by the sentence on lines 85 – 87.

2. I was confused about the simulation setups. For instance, it appeared like the simulations were performed and compared versus observations at a point, although 1) the model is often used for distributed simulations, and 2) the use of snow/vegetation fractions implied a gridcell or patch of much larger size. While the ISBA parameters (transmission coefficient, veg parameter, etc.) were left unchanged or defined by the datasets in Table 2, the MEB canopy longwave radiation transmission was tuned. Although the tuned radiation transmission (0.4) was close to the default transmission (0.5), the snow depth RMSE for the default transmission was larger (Figure 11) and approached the default ISBA simulations (Table 8). It would be nice to include the MEB simulation using all default parameters in a comparison (maybe similar to what was done for the simulation with no interception in Figure 6).

3. The model developments are valuable for not only offline land surface models. I was therefore curious to hear more about how the authors expect the ISBA MEB developments to influence coupled and distributed land-atmosphere simulations. Also, how do you expect the model developments to perform or differ in landscapes (such as the

[Figure]

United States Pacific Northwest Cascades) where elevation gradients are large, temperatures are warmer, snow depth is typically deeper, and the canopy intercepts much larger amounts of snow for longer portions of the snow season? A brief discussion about model transferability would be valuable. Finally, from a modeling perspective, how much (if any) do the MEB developments increase the computational cost?

Technical comments:

Line 2 "...adopts a default configuration...": Does this mean the default ISBA model? Maybe consider simplifying this whole sentence to clearly state that the ISBA model uses a composite soil-vegetation energy budget that struggles with representing snow in forested regions.

Lines 11-12 "A consistent positive impact for soil temperatures...": With the statistics that follow (-6.2 to -0.1 K), I am not sure if "consistent positive impact" means that the soil temperature always increases (positively), or if simulated soil temperature improves.

Line 14 "...time of ablation...": Does this mean the date of first snow-absence (or melt-out), or the rate at which snow melts? You use "last day of snow" in the results. For consistency, I would pick one and stick with it.

Line 16: "cause" should be "caused".

Lines 20 – 21 "...one third of which consists of boreal forest which corresponds to subarctic and cold climates.": Maybe revise for conciseness to "...one third of which consists of boreal forests in subarctic and cold climates".

Line 52: "2009, Rutter et al." should be "Rutter et al., 2009".

Line 55 – 56 "...they determined that liquid water retention was a key process required for simulating the accurate timing and amount of snowmelt and thus discharge": Can you be more specific? After reading the Boone et al. (2004) paper, I am still not sure what you mean here. I am guessing that liquid water retention references the soil

column and that soil columns with a larger holding capacity simulated daily discharge better. However, the composite ground representation seemed to be a first-order driver of whether snowmelt was even entering the soil column at the correct time. By "liquid water retention" do you mean delayed snowmelt by non-composite snow schemes?

Line 62 "...certain snow processes...": I would be explicit here (interception, solar shading, longwave enhancement, etc.). What processes require "explicit representation of the vegetation canopy"? Also, what does "explicit" mean (canopy height, canopy density, subgrid canopy coverage/placement, vegetation species, LAI, etc.)? The required information about the canopy vary across different models.

Line 69: "GGMs" should be "GCMs".

Line 83: "computations" is misspelled.

Line 101 "...certain key features": Can you be explicit here (snow depth, SWE, etc.)?

Line 116: I would delete "for research studies which consists in" and put the colon after "...default ISBA configuration, where:"

Line 137: Some models partition snow layers based on SWE instead of snow depth. It is worth mentioning what ISBA does here.

Section 2.1.4: I think the discussion at the end of the section highlights one of the most important model developments. This is alluded to briefly in the abstract, in the parenthetical from lines 147 – 148, and lines 165 – 166. Sections 2.1 and 2.2 demonstrate the differences between these two model developments (ISBA and MEB) well. However, the impact on heat/energy fluxes (Figure 3) is already demonstrated in Napoly et al. (2017). I think these results could be referenced early-on in this manuscript and used to elaborate how these changes are important for this snow-modeling investigation here. I also think that Figure 2 is a great conceptual that could be referenced earlier to demonstrate how the models differ in their subgrid representation of snowpack.

Lines 258 – 262: I have concerns about the assumption that intercepted snow has negligible effect on the canopy albedo. Although Pomeroy and Dion (1996) found canopy structure and solar angles to be the first-order drivers of radiation absorption by the canopy, multiple studies since have linked differences between observed and modeled albedo (and differences between models) to modeled canopy interception (e.g., Bartlett et al., 2006; Loranty et al., 2014; Roesch and Roeckner, 2006; Thackeray et al., 2014). What sort of impact do you anticipate if the canopy albedo were to vary with interception? Canopy typically intercepts much more snowfall in the United States Pacific Northwest and many other maritime snow climates. Therefore, how do you expect this assumption to influence simulations in other climates?

Lines 270-271: At this point of the text, I want to know more about the canopy interception model and parameters that are used. I think it is first mentioned in Line 482 that you use the Hedstrom and Pomeroy (1998) method. I think the Hedstrom and Pomeroy parameterization is a good choice since it was developed for this particular region. However, it is worth noting that the Hedstrom and Pomeroy method varies dramatically from the Storck et al. (2002) method which does better for regions with warmer, and more cohesive snowpack. In fact, a number of snow interception parameterizations exist (e.g., Hedstrom and Pomeroy, 1998; Roesch et al., 2001; Storck et al., 2002), most of which are heavily-parameterized and are not very transferable between climates. I really like the discussion on snow interception sensitivity in Section 4.2.4. I think a simple 1-2 sentence acknowledgement about different interception routines and the impact of tuning interception parameters on modeled snowpack would be valuable.

Line 288: Delete "In order" or make "To" lower-case.

Section 3: It would be nice to know what resolution these simulations were being performed at since measurements are at points. How co-located in space are observations? Do you expect any variability from ground measurements like manual SWE measurements which likely did not come from the same spot each time? I especially find the SWE measurements in OBS and OJP, and the accuracy versus the simulations, interesting in water-year 2004 (Figure 5).

Line 355 – 357: Is the difference in sublimation averaged across all model domains? I would expect the ratio between in-, and under-canopy sublimation to be different as site characteristics vary. In fact, in lines 362 – 363, it looks like it does vary across the sites. What is the ratio between "sublimation of the snowpack and total sublimation of snow" (lines 362 – 363)? Is this the ratio between the snow sublimated from the ground layer versus the total (sublimated from the ground and canopy)? If so, the average ratio here (0.45) seems to align with your 12%:27% split presented in the topic sentence.

Line 365: I think Figure 6 is referenced before Figure 5.

Line 367: For consistency, "Table 5" should be "Tab. 5".

Lines 380 – 385: Is RMSE calculated only for periods where snow exists in 1) the observation? 2) either the observation or simulation? or 3) for the full 3-year period including snow-absence? I think this may have been answered in lines 430 – 440. If so, please move this up earlier.

Line 500: Change "has" to "was" or "has been".

Figure 1: There is no caption for this Figure. The layout for this figure may also be difficult for typesetting. Could these figures go beside each other with labels specifying the ISBA and ISBA-MEB frameworks?

Figure 5 and Figure 9: The horizontal time axis represents an explicit date (as compared to an annual composite or average). Can "Time (Year)" be changed to an explicit date (e.g., Jul 01 through Jul 04)?

There are no references to Table 6 or Table 7 in the text.

Figures 7, 8, 10, and 11. Label subplots (a, b, c, etc.) in accordance to references in the text and figure captions.

References:

Bartlett, P.A., MacKay, M.D., Verseghy, D.L., 2006. Modified snow algorithms in the

Canadian land surface scheme: Model runs and sensitivity analysis at three boreal forest stands. Atmosphere-Ocean 44, 207–222. https://doi.org/10.3137/ao.440301

Hedstrom, N.R., Pomeroy, J.W., 1998. Measurements and modelling of snow interception in the boreal forest. Hydrological Processes 12, 1611–1625. https://doi.org/10.1002/(SICI)1099-1085(199808/09)12:10/11<1611::AID-HYP684>3.0.CO;2-4

Loranty, M.M., Berner, L.T., Goetz, S.J., Jin, Y., Randerson, J.T., 2014. Vegetation controls on northern high latitude snow-albedo feedback: observations and CMIP5 model simulations. Global Change Biology 20, 594–606. https://doi.org/10.1111/gcb.12391

Napoly, A., Boone, A., Samuelsson, P., Gollvik, S., Martin, E., Seferian, R., Carrer, D., Decharme, B., Jarlan, L., 2017. The interactions between soil-biosphere-atmosphere (ISBA) land surface model multi-energy balance (MEB) option in SURFEXv8 - Part 2: Introduction of a litter formulation and model evaluation for local-scale forest sites. Geoscientific Model Development 10, 1621–1644. https://doi.org/10.5194/gmd-10-1621-2017

Pomeroy, J.W., Dion, K., 1996. Winter Radiation Extinction and Reflection in a Boreal Pine Canopy: Measurements and Modelling. Hydrological Processes 10, 1591–1608. https://doi.org/10.1002/(SICI)1099-1085(199612)10:12<1591::AID-HYP503>3.0.CO;2-8

Roesch, A., Roeckner, E., 2006. Assessment of Snow Cover and Surface Albedo in the ECHAM5 General Circulation Model. Journal of Climate 19, 3828–3843.

Roesch, A., Wild, M., Gilgen, H., Ohmura, A., 2001. A new snow cover fraction parametrization for the ECHAM4 GCM. Climate Dynamics 17, 933–946. https://doi.org/10.1007/s003820100153 Storck, P., Lettenmaier, D.P., Bolton, S.M., 2002. Measurement of snow interception and canopy effects on snow accumulation and melt in a mountainous maritime climate, Oregon, United States. Water Resources

[Figure]

Research 38, 5-1-5–16. https://doi.org/10.1029/2002WR001281

Thackeray, C.W., Fletcher, C.G., Derksen, C., 2014. The influence of canopy snow parameterizations on snow albedo feedback in boreal forest regions. Journal of Geophysical Research: Atmospheres 119, 9810–9821. https://doi.org/10.1002/2014JD021858
* * *

---

## Referee Comment (RC2) · Anonymous Referee #2 · 9 Sep 2020

This study presents the evaluation of the ISBA-MEB snow model in 3 forest sites in Canada. Despite being based only on 3 stations, these are well known and with good quality observations that allow a detailed evaluation as presented in this study. A sensitivity analysis is also performed and identifies 1 important parameters of the MEB scheme which is relevant for the snow simulations. This is a relevant study for the snow community as it highlights several processes important in the modeling of snow and soil conditions in forest areas. It is also relevant for a wider climate community due to the role of these areas in the response to a warmer climate. The manuscript is well organized and clear. I only found a few details, listed below, that require some attention from the authors.

Fig.1 legend is missing

Fig.2 Which snow density was assumed ? Snow cover fraction is a function of snow depth (D, m) and not Snow Water equivalent. Please indicate which density is used, or plot snow fraction as function of snow depth.

Line 288: "In order To" : "In order to"

Line 367: Defining last day of snow when SND<0.2m and below that for the following two weeks. The mean annual cycle of snow depth in Figure 6 shows that ISBA simulations on average never reach 20cm of snow depth in the OAS site. In years when SND is always < 0.2 how does this identification of last day of snow works in a simulation ? A value of 0.1 seems more reasonable. Would changing from 20 cm to 10 cm change significantly the metrics in Table 5 ?

Line 377; "Also, Fig. 5 seems to indicate that the snow density is well modeled since underestimation or overestimation of SND and SWE are consistent for both models." This is true for OAS and OJP, but the OBS results in year 2 and 3 (Fig 5) indicate a reasonable performance of snow depth but a large underestimation of snow mass in year 2 and over-estimation in year 3 (also OJP in year 3). Could this be related with snow density errors linked with different winter conditions between year 2 and 3?

Fig 7. Missing panel names (a,b,c) which are used in the text (e.g. line 399). The 3rd datetick seems wrong "03/29/2006" should be "03/28/2006" ?

Line 437: Suggest to remove "somewhat"

---

## Author Comment (AC1) · 20 Oct 2020

1. I liked the organization of this manuscript. The results were presented in a fashion that made both the model developments and findings easy to understand. On my second readthrough, I found that my mind was already primed to identify the issues with the default ISBA model that made modeling snow in forests troublesome. However, when I first read through this manuscript, I wanted to know more about the sources of model errors before getting into the model details. Only in lines 195 – 200, and the following sections describing MEB, did I start to understand what was being corrected. Specifics about the changes to the model framework and how that influenced the snowpack could be put earlier to prime the reader for what to expect. This need not be lengthy (only a few sentences) and could be included in an individual paragraph,

headed by the sentence on lines 85 – 87.

We have added a few lines after lines 85-87 to enhance the description of certain weaknesses of the ISBA composite scheme to model the snow pack and to improve the transition to the description of MEB. Indeed, we feel the transition is a bit more smooth now.

2. I was confused about the simulation setups. For instance, it appeared like the simulations were performed and compared versus observations at a point, although 1) the model is often used for distributed simulations, and 2) the use of snow/vegetation fractions implied a gridcell or patch of much larger size.

In the current study, the model is used at the local scale (1d simulation) which is assumed to have a length-scale in the order of about 10 to 100m, but of course this is somewhat arbitrary. But when the model is used at such small ("local") scales, the sub-grid heterogeneity parameterizations (used at larger scales) collapse to correspond to a single land cover type and homogeneous hydrological fluxes (such as infiltration, runoff etc). For local case studies, the model input parameters (Tab.2) are defined to corresponds as much as possible to the study sites which are supposed to be homogeneous over that scale. In addition, the observations (notably the turbulent fluxes) are assumed to be applicable to this scale. Finally, in spatially distributed applications, with length-scales (grid cells) generally ranging from 102 to 105 m (mesoscale meteorological and hydrological applications, to climate modeling), the model input parameters are aggregated (up-scaled) using the fine scale data from ECOCLIMAP to the chosen resolution thus accounting for sub-grid heterogeneity in a relatively simplified but economical manner (Noilhan and Lacarrerre, 1996). A sentence is added at the beginning of paragraph 4, line 326.

3. While the ISBA parameters (transmission coefficient, veg parameter, etc.) were left unchanged or defined by the datasets in Table 2, the MEB canopy longwave radiation transmission was tuned. Although the tuned radiation transmission (0.4) was close to

the default transmission (0.5), the snow depth RMSE for the default transmission was larger (Figure 11) and approached the default ISBA simulations (Table 8). It would be nice to include the MEB simulation using all default parameters in a comparison (maybe similar to what was done for the simulation with no interception in Figure 6).

There is a misunderstanding in the RMSE we calculated, thus we have attempted to make this more clear and consistent in accordance with this reviewer. The RMSE from Table 8 corresponds to a calculation made with the entire data set, meaning including snow-free periods. The RMSE from the sensitivity test figure corresponds to data for which snow cover is present (in the measurements). For this reason, the second RMSE values are higher. We modified the table for consistency using RMSE calculated with snow present on the ground. You can now see that RMSE from the table corresponds to the figure and that those calculated with the ISBA model are much higher than the one using both tau=0.4 or tau=0.5, which are very close to each other.

For the reviewer, we have made an additional figure (but ti has not been put in the paper...but could be if the reviewer or editor feel it is necessary). The figure RC1,1 shows the composite annual cycle of snow depth for the simulations using MEB performed with both the 0.4 and 0.5 value of the tau coefficient. It can be seen that this does not strongly influence the modeled snowpack, at least compared to the change between ISBA and MEB. Because this coefficient was at first set as a default value in Boone et al. 2017 without any evaluation, we chose here to change that value to 0.4

4. The model developments are valuable for not only offline land surface models. I was therefore curious to hear more about how the authors expect the ISBA MEB developments to influence coupled and distributed land-atmosphere simulations.

Indeed, as mentioned in the response to comment 2, the next step for model evaluation is currently underway. Now that the model has been shown to correct certain significant systematic biases (soil temperature, snow ablation timing. . ..) at well-documented and rather representative Boreal forest sites for different tree species and characteris-

tics, and also in preliminary offline global scale evaluations using standard reanalysis products over long time periods. Such simulations will be evaluated using observed permafrost depths and spatial distributions, snow cover fraction (satellite based), point soil temperature measurements over high latitude local scale sites in forested areas, and river discharge. Preliminary work was done over France by Napoly et al. (2017), and evaluations using the updated distributed hydrological system over France are to begin soon as mentioned in the perspectives of Le Moigne et al (2020), and work is beginning at the global scale in a similar manner as presented by Decharme et al. (2019), who also mentions in their persepctives that MEB will be used. Once those steps are finished, the next step will be testing in the fully-coupled CNRM-CM climate model. The model is also currently being tested within the context of operational NWP within the HIRLAM consortium. We have added test to Lines 588-599 which refer to these perspectives.

5.Also, how do you expect the model developments to perform or differ in landscapes (such as the United States Pacific Northwest Cascades) where elevation gradients are large, temperatures are warmer, snow depth is typically deeper, and the canopy intercepts much larger amounts of snow for longer portions of the snow season? A brief discussion about model transferability would be valuable.

Indeed this is a good point. The motivation for MEB development was initially mainly for snowy forested regions. But, the focus was on Siberian and high latitude forests since systematic near surface/surface cold biases were identified for those regions in both offline (see for example, Decharme et. al., 2019) and coupled (climate) runs. Also, there was a motivation to perform a detailed study using the Berms data owing to it's quality, visibility (recently as a part of the ESM-SnowMIP model inter-comparison study) and the fact that these sites most readily addressed the problems mentioned previously. But indeed, there is work going on at CEN-Météo-France on MEB coupled to the detailed snow process model CROCUS (and the ES scheme, used herein, in parallel) for the new extension of the Col de Porte site to an adjoining forest. This is

a relatively (compared to Boreal regions) warm and wet climate, rather typical of the Alps. And, work to evaluate MEB for other forested sites is also planned, but for the current study, the main motivation was to address the areas/type of climate/cover for which MEB brings in the most pressing and dramatic impact. We specified the choice of the Berms sites in the introduction (l.100-101). Also, we added a sentence at the end of the conclusion to mention this future (Col de Porte site) work, along with references.

6.Finally, from a modeling perspective, how much (if any) do the MEB developments increase the computational cost

When running SURFEX for a standard single point run (for the runs within the current study) using the default GNU gfortran compile options, the MEB option increases the run time by 5.8Âă% But note that in coupled runs, this additional cost becomes quite small as the surface is relatively inexpensive compared to the atmosphere (the surface is generally a fewÂă% of run time in our systems). Thus the cost of adding MEB should relatively insignificant. We added a short discussion of this to line 222.

7. Line 2 "...adopts a default configuration...": Does this mean the default ISBA model? Maybe consider simplifying this whole sentence to clearly state that the ISBA model uses a composite soil-vegetation energy budget that struggles with representing snow in forested regions.

We simplified the sentence following your advice.

8. Lines 11-12 "A consistent positive impact for soil temperatures...": With the statistics that follow (-6.2 to -0.1 K), I am not sure if "consistent positive impact" means that the soil temperature always increases (positively), or if simulated soil temperature improves.

We rephrased the sentence to better explain that the improvement of soil temperatures is consistent with the improvement of the ground heat flux.

9. Line 14 "...time of ablation...": Does this mean the date of first snow-absence (or

melt-out), or the rate at which snow melts? You use "last day of snow" in the results. For consistency, I would pick one and stick with it.

We decided to keep here the expression 'last day of snow' as it refers to the score that is later used in the manuscript.

10. Line 16: "cause" should be "caused".

corrected

11. Lines 20 – 21 "...one third of which consists of boreal forest which corresponds to subarctic and cold climates.": Maybe revise for conciseness to "...one third of which consists of boreal forests in subarctic and cold climates".

Thank you: we have done this (for simplification).

12. Line 52: "2009, Rutter et al." should be "Rutter et al., 2009".

corrected

13. Line 55 – 56 "...they determined that liquid water retention was a key process required for simulating the accurate timing and amount of snowmelt and thus discharge": Can you be more specific? After reading the Boone et al. (2004) paper, I am still not sure what you mean here. I am guessing that liquid water retention references the soil column and that soil columns with a larger holding capacity simulated daily discharge better. However, the composite ground representation seemed to be a first-order driver of whether snowmelt was even entering the soil column at the correct time. By "liquid water retention" do you mean delayed snowmelt by non-composite snow schemes?

We have restructured the text to be more clear (line 55). Indeed, an important result of Boone et al. (2004) was to show that the retention of liquid water (owing to a storage capacity in the snow and also possible refreezing of this liquid water) improved both the predicted peak discharge and phase/timing of this peak for the high altitude Durance basin, mainly owing to the increased SWE and the fact that melt events or even

rainfall on the snowpack did not result in instantaneous runoff. Indeed, soil processes could also play a role, but the participant LSMs were requested to use the prescribed soil parameters and depths in Rhone-AGG and soils were relatively thin in mountain regions thus diminishing the residence time (so differences in soil processes could indeed contribute a bit, but through our experience over this basin and based on the similar behavior by LSMs with similar snow schemes, we came to this conclusion that it was the liquid water retention and refreezing processes which were the most critical). Only schemes with this process explicitly modeled were able to simulate a reasonable discharge for this basin (of course now, many LSMs include this, but at that time, there were relatively few).

14. Line 62 "...certain snow processes...": I would be explicit here (interception, solar-shading, longwave enhancement, etc.). What processes require "explicit representation of the vegetation canopy"? Also, what does "explicit" mean (canopy height, canopy density, subgrid canopy coverage/placement, vegetation species, LAI, etc.)? The required information about the canopy vary across different models.

We modified the text by specifying the key physical processes which are difficult to take into account in composite soil-vegetation scheme such as the ISBA model (line 62). 'Explicit' means that the model considers distinct layers for the ground and the canopy. In the composite ISBA model, there is only one layer which has the characteristics (albedo, roughness, emissivity ..) of both the ground and the vegetation (these characteristics correspond to the weighted average of the characteristics of each surface type-soil and vegetation-based on a estimated vegetation fraction). A sentence is added to explain this more clearly (line 64).

15. Line 69: "GGMs" should be "GCMs".

corrected

16. Line 83: "computations" is misspelled.

corrected

17. Line 101 "...certain key features": Can you be explicit here (snow depth, SWE, etc.)?

We added examples in the text to be more explicit about two aspects that are investigated in the study.

18. Line 116: I would delete "for research studies which consists in" and put the colon after"...default ISBA configuration, where:"

The text is modified to follow this comment.

19. Line 137: Some models partition snow layers based on SWE instead of snow depth. It is worth mentioning what ISBA does here.

ISBA partitions the layers based on snow depth. At the start of each time step, new snowfall or canopy unloaded snow is incorporated into the snowpack. Then, the grid is recomputed based on the total snow depth following a set of rules to provide the best vertical resolution at both the snow-atmosphere and snow-soil interfaces (the reasoning is discussed in detail in Decharme et al., 2019). Snow properties between layers are then potentially blended to assure total snowpack enthalpy and mass conservation during the grid reset. We slightly adapted the description of ISBA-ES in section 2.1 to mention this more clearly.

20. Section 2.1.4: I think the discussion at the end of the section highlights one of the most important model developments. This is alluded to briefly in the abstract, in the parenthetical from lines 147 – 148, and lines 165 – 166. Sections 2.1 and 2.2 demonstrate the differences between these two model developments (ISBA and MEB) well. However, the impact on heat/energy fluxes (Figure 3) is already demonstrated in Napoly et al. (2017). I think these results could be referenced early-on in this manuscript and used to elaborate how these changes are important for this snow-modeling investigation here. I also think that Figure 2 is a great conceptual that could be referenced earlier

to demonstrate how the models differ in their subgrid representation of snowpack.

Results of Napoly et al. (2017) are now referenced at lines 100-101. It is true that the impact of MEB and litter on fluxes are already demonstrated in Napoly et al. (2017) and that Fig. 3 might seem redundant, but we think that it is beneficial to the reader to start the analysis with a global view of the results. Following you comment, we cited Figure 2 earlier in the text (lines 123-125) and insisted that the snow fraction parametrization is a key aspect of the limitation of the ISBA model: MEB has permitted us to remove a highly empirical (and not very physical) parameterization compared to the composite scheme (the composite vegetation snow cover fraction).

21. Lines 258 – 262: I have concerns about the assumption that intercepted snow has negligible effect on the canopy albedo. Although Pomeroy and Dion (1996) found canopy structure and solar angles to be the first-order drivers of radiation absorption by the canopy, multiple studies since have linked differences between observed and modeled albedo (and differences between models) to modeled canopy interception (e.g., Bartlettet al., 2006; Loranty et al., 2014; Roesch and Roeckner, 2006; Thack-eray et al., 2014).What sort of impact do you anticipate if the canopy albedo were to vary with interception? Canopy typically intercepts much more snowfall in the United States PacificNorthwest and many other maritime snow climates. Therefore, how do you expect this assumption to influence simulations in other climates?

Indeed it outwardly appears like it is a strong assumption to neglect the effect of inter-cepted snow on the albedo. However, we think that it is reasonable using the following arguments:

• in general, the canopy albedo during the winter season is not critical since solar radiation is relatively low so that the effective albedo considered by the model is less significant for the energy budget calculation. • in spring, when solar radiation is higher, snow events tend to become more scarce and the unloading + melting parame-terizations get rid of intercepted snow quite effectively/rapidly and prevents interception

fractions to approach unity, at least for any extended time period.

However, in order to test the above assumptions, we implemented in the code this effect following Roesch and Roeckner (2006). Using their value of 0.2 for the so called "albedo of the snow covered part of the canopy", we found no effect on the snow pack. As an extreme academic test, we made additional tests using 0.9 (which is probably too excessive/large): when using the 0.9 value we found differences of a few millimeters in SWE maximum for a given day for the 3 sites, which is quite small. The impact on ablation timing was negligible. As a conclusion, on average over the snow season we think this assumption is valid. But indeed, for numerical weather prediction which focuses on the short term, it might however be interesting to consider this effect in the future, but more study will be required. In fact, interception and unloading etc. processes are the focus of the now cited work at Col de Porte (Helbig et al., 2020, line 596), for which MEB will be extensively tested. We have added lines 281-283.

22. Lines 270-271: At this point of the text, I want to know more about the canopy interception model and parameters that are used. I think it is first mentioned in Line 482 that you use the Hedstrom and Pomeroy (1998) method. I think the Hedstrom and Pomeroy parameterization is a good choice since it was developed for this particular region. However, it is worth noting that the Hedstrom and Pomeroy method varies dramatically from the Storck et al. (2002) method which does better for regions with warmer, and more cohesive snowpack. In fact, a number of snow interception parameterizations exist (e.g., Hedstrom and Pomeroy, 1998; Roesch et al., 2001; Storck et al.,2002), most of which are heavily-parameterized and are not very transferable between climates. I really like the discussion on snow interception sensitivity in Section 4.2.4. I think a simple 1-2 sentence acknowledgement about different interception routines and the impact of tuning interception parameters on modeled snowpack would be valuable.

We added a sentence in section 4.2.4 (l.492-493) to mention another parameterization. The interception of the snow is indeed an important process that has now been added

owing to MEB among several others (more realistic radiative transfer, within canopy turbulence), but we have found that the most significant impact obtained with MEB for snow covered forests is that we are able to eliminate a highly empirical and conceptual representation of snow interception (in the composite scheme) which also permits an explicit representation of snow on the forest floor. This has a significant impact on the ablation timing. But indeed, we plan to examine the interception parameterization in more detail, namely owing to work in progress now mentioned in the Perspectives at the Col de Porte site (again, the Helbig et al. 2020 reference has been added in this vein).

23. Line 288: Delete "In order" or make "To" lower-case.

Corrected

24. Section 3: It would be nice to know what resolution these simulations were being performed at since measurements are at points.

Simulations are 1d so that the notion of resolution does not really explicitly exist in such case. However, indeed, the observations are assumed to be valid over a certain spatial scale (notably the turbulent fluxes). To perform the simulations, we assume that the land cover where the observations were made is homogeneous over a large area (as discussed in our response to this reviewer's comment 2), so that the impact of any other different land cover around the observation point should be relatively small. The idea is that the land cover should be fairly homogeneous over the approximate footprint of the turbulence measurements also which can be time varying and rather complex in itself; Cuxart and Boone, 2020 (BLM) give a discussion on this aspect...eddy covariance observations and the corresponding footprint/scale: but of course many papers discuss this issue, the aforementioned reference lists many of those studies. We modified the first sentence of section 4 to be more explicit on that point.

25. How colocated in space are observations? Do you expect any variability from ground measurements like manual SWE measurements which likely did not come from

the same spot each time? I especially find the SWE measurements in OBS and OJP, and the accuracy versus the simulations, interesting in water-year 2004 (Figure 5).

Indeed this is right, there is variability on snow depth measurements. In the current study, we used the data called 'UC' for 'under canopy' as : • it was the only available at one of the three sites. This was the case for the OBS site for which there were measurements at 3 locations at the site, but measurements were not available for the full period. • we assume that UC is more relevant for model comparison with the model than other measurements such as : 'canopy gap' measurements.

For the OBS site, measurements called 'North West' (NW) and 'North East' (NE) were available in addition to the UC data (see Figure RC1,2). SND was higher at these locations than at the UC location. However, in our case of MEB evaluation considering the inter-site SWE variability does not change the conclusion for 2 reasons: we have computed the SND statistical metrics for ISBA and MEB using the other measurements (i.e. not UC) and the results degrade slightly with about the same magnitude for both models. But perhaps more importantly, as the improvement due to the MEB option would remain the same because the average ablation date for the 3 obs only differs by 5 days maximum among the 3 measurements, and MEB improves the ablation by 2-3 weeks compared to ISBA (so we obtain the same dramatic improvement in ablation timing regardless of which of the 3 measures of SND we use).

26. Line 355 – 357: Is the difference in sublimation averaged across all model domains?

We hope that the previous answer should now clarify/answer this question. As explained, there is no notion of domain or area here as simulations are 1d. The sublimation calculated by the model corresponds to a quantity per square meter over the scale of a so-called parcel/field.

27. I would expect the ratio between in-, and under-canopy sublimation to be different as site characteristics vary. In fact, in lines 362 – 363, it looks like it does vary across

the sites.

Indeed, it varies across the sites from 0.42 to 0.51 as mentioned line 379.

28. What is the ratio between "sublimation of the snowpack and total sublimation of snow" (lines 362 – 363)? Is this the ratio between the snow sublimated from the groundlayer versus the total (sublimated from the ground and canopy)?

Yes

29. If so, the average ratio here (0.45) seems to align with your 12%:27% split presented in the topic sentence.

Yes exactly, we found it matches fairly well with values (we) found in the literature (0.45). This is why we used it here and calculated the corresponding value for each site (0.42, 0.45, 0.51).

30. Line 365: I think Figure 6 is referenced before Figure 5.

You are right, the names of the figures have been exchanged.

31. Line 367: For consistency, "Table 5" should be "Tab. 5".

Corrected.

32. Lines 380 – 385: Is RMSE calculated only for periods where snow exists in 1) the observation? 2) either the observation or simulation? or 3) for the full 3-year period including snow-absence? I think this may have been answered in lines 430 – 440. If so, please move this up earlier.

The RMSE from the table have been modified and are now calculated using the data which correspond to a presence of snow in the observations. The legend of Tab. 8 is modified to avoid any doubt, also the sentence line 380 is rephrased to reflect that the score refers to the whole period and not the 3 years as was stated in the first manuscript version.

33. Line 500: Change "has" to "was" or "has been".

Corrected

34. Figure 1: There is no caption for this Figure. The layout for this figure may also be difficult for typesetting. Could these figures go beside each other with labels specifying the ISBA and ISBA-MEB frameworks?

The legend was hidden, this is now corrected. Also, the figures are now beside each other to follow the comment.

35. Figure 5 and Figure 9: The horizontal time axis represents an explicit date (as compared to an annual composite or average). Can "Time (Year)" be changed to an explicit date (e.g., Jul 01 through Jul 04)?

The explcit dates have been added to follow your recommendation in Figures 5 (now 6) and 9.

36. There are no references to Table 6 or Table 7 in the text.

References have been added in section 4.1.3

37. Figures 7, 8, 10, and 11. Label subplots (a, b, c, etc.) in accordance to references in the text and figure captions.

We added the labels on Figure 9 and 7 as we refer to it in the text. We left the two others unchanged and adapted the text to be consistent.

———————————————

[Figure]

Fig. 1.

[Figure]

**Fig. 2.**

---

## Author Comment (AC2) · 20 Oct 2020

The legend was hidden, this is now corrected

Fig.2 Which snow density was assumed ? Snow cover fraction is a function of snowdepth (D, m) and not Snow Water equivalent. Please indicate which density is used, or plot snow fraction as function of snow depth.

This is correct: the legend has been adapted to clearly mention that we considered a density of 200 kg m-3 for the figure.

Line 288: "In order To" : "In order to"

Corrected

[Figure]

Line 367: Defining last day of snow when SND<0.2m and below that for the following two weeks. The mean annual cycle of snow depth in Figure 6 shows that ISBA simulations on average never reach 20cm of snow depth in the OAS site. In years when SNDis always < 0.2 how does this identification of last day of snow works in a simulation ?A value of 0.1 seems more reasonable. Would changing from 20 cm to 10 cm change significantly the metrics in Table 5 ?

There is a mistake here, the threshold value used is 2cm and not 20cm as we wrote in the text. We tested higher values (3, 4 and 5 cm) and the results didn't really change the metrics of table 5. We apologize for this confusion and thank the reviewer for spotting this typographical error.

Line 377; "Also, Fig. 5 seems to indicate that the snow density is well modeled since underestimation or overestimation of SND and SWE are consistent for both models."This is true for OAS and OJP, but the OBS results in year 2 and 3 (Fig 5) indicate a reasonable performance of snow depth but a large underestimation of snow mass in year 2 and over-estimation in year 3 (also OJP in year 3). Could this be related with snow density errors linked with different winter conditions between year 2 and 3?

Indeed, this sentence is less relevant for years 2 and 3 at the OBS site, even if over-estimation and under-estimation between SND and SWE are consistent. We added a sentence to soften the remark. The main goal of this response is essentially to point out that the weakness of the ISBA model or the improvement of the MEB model cannot be linked to a density issue that would come from the snow model. We added a sentence to specify that more data would be necessary to accurately validate the SWE or modeled snow density (line 397).

Fig 7. Missing panel names (a,b,c) which are used in the text (e.g. line 399). The 3rd datetick seems wrong "03/29/2006" should be "03/28/2006" ?

Corrected

Line 437: Suggest to remove "somewhat"

Corrected
* * *